# Envy-Free Allocation of Indivisible Goods via Noisy Queries

**Zihan Li** [1]    **Yan Hao Ling** [2]    **Jonathan Scarlett** [3]    **Warut Suksompong** [3]

## Abstract

We introduce a problem of fairly allocating indivisible goods (items) in which the agents' valuations cannot be observed directly, but instead can only be accessed via noisy queries. In the two-agent setting with Gaussian noise and bounded valuations, we derive upper and lower bounds on the required number of queries for finding an envy-free allocation in terms of the number of items, $m$, and the negative-envy of the optimal allocation, $\Delta$. In particular, when $\Delta$ is not too small (namely, $\Delta \gg m^{1/4}$), we establish that the optimal number of queries scales as $\frac{\sqrt{m}}{(\Delta/m)^2} = \frac{m^{2.5}}{\Delta^2}$ up to logarithmic factors. Our upper bound is based on non-adaptive queries and a simple thresholding-based allocation algorithm that runs in polynomial time, while our lower bound holds even under adaptive queries and arbitrary computation time.

## 1. Introduction

The fair allocation of indivisible goods (henceforth referred to as "items") is a fundamental problem at the intersection of computer science and economics with diverse applications, and has been studied extensively in recent years. This topic has been investigated under a wide variety of fairness criteria, each having its own benefits and implications; for some recent surveys, see (Moulin, 2019; Aziz, 2020; Walsh, 2020; Suksompong, 2021; Amanatidis et al., 2023). Among these fairness notions, *envy-freeness* (and its variations) is among the most fundamental and widely-studied. In an envy-free allocation, each agent values the set of items they receive at least as much as they value the items given to any other single agent. Since envy-freeness can be impossible

to satisfy (e.g., when there is only one item) or computationally hard to achieve, it is common to study relaxations such as *envy-freeness up to one item* (EF1).

A prevailing assumption in the literature is that the valuations of the agents are known exactly, or can be queried to learn the exact value. In this paper, we are interested in scenarios where such queries may be *noisy*, motivated by scenarios such as the following:

(i) The "utilities" are not measurable or human-chosen, but instead need to be estimated via simulations that are stochastic in nature (e.g., estimating how much a particular resource will help each team in a company's internal division problem).

(ii) Each so-called "agent" is actually a group of people, and the "utility" is considered to be an average over the group members. To query an item, we may take a random group member and ask for their valuation.[1]

(iii) More generally, noise may arise in settings where agents make mistakes in their evaluations, the evaluations fluctuate over time, and so on.

It is likely highly challenging to devise a general noise model capturing all such scenarios, so as a natural starting point, we focus on the fundamental model of additive Gaussian noise; see Remark 1 in Section 2 for a discussion on our noise model's justification and limitations. This model of noisy querying matches that of the extensively studied multi-armed bandit problem (Lattimore & Szepesvári, 2020), and also falls under the general topic of algorithms with noisy queries (e.g., see (Feige et al., 1994; Addanki et al., 2021; Zhu et al., 2023)), though the latter typically involves discrete noise models (e.g., binary-output queries that get flipped with some probability).

As we will see, the presence of noise leads to an inherently distinct problem that is technically complex, even in our restricted setting of two agents, additive utilities,[2] and

---

[1]Meta Singapore (work done prior to joining Meta) [2]Nanyang Technological University, Singapore [3]National University of Singapore, Singapore. Correspondence to: Zihan Li <zihan@meta.com>, Yan Hao Ling <yanhao.ling@ntu.edu.sg>, Jonathan Scarlett <scarlett@comp.nus.edu.sg>, Warut Suksompong <warut@comp.nus.edu.sg>.

*Proceedings of the 43rd International Conference on Machine Learning*, Seoul, South Korea. PMLR 306, 2026. Copyright 2026 by the author(s).

---

[1]This formulation assumes that fairness is only required on an averaged group level, not an individual level.

[2]The two-agent setting is fundamental in fair division, with many prominent works focusing on this setting. Moreover, several fair division applications such as divorce settlement and interna-

Gaussian noise. We show that the number of queries required for finding an envy-free allocation depends crucially on the "optimal negative envy" $\Delta > 0$ (or a lower bound thereof), and grows to $\infty$ as $\Delta \to 0$. Our main results reveal that with $m$ items having valuations in $[0, 1]$ and a constant noise variance, the optimal query complexity is[3] $\widetilde{O}\left(\frac{m^{2.5}}{\Delta^2}\right)$ in broad regimes of $\Delta$ as $m \to \infty$. Before summarizing our contributions in more detail, we discuss related prior work.

### 1.1. Related Work

To the best of our knowledge, our problem setup has not been considered before, and there are no existing results that are directly comparable to ours. Nevertheless, we proceed to give an overview of some of the most related existing literature. We subsequently use the terminology *utility* and *valuation* interchangeably to mean how much an agent values a given set of items. For the purpose of our discussion, it suffices to note the following fairness notions:

- Envy-free (EF): Each agent prefers their own bundle (i.e., set of received items) to that of any other agent;

- Envy-free up to one item (EF1): Each agent prefers their own bundle to that of any other agent after removing *a specific* item from the latter;

- Envy-free up to any item (EFX): Each agent prefers their own bundle to that of any other agent after removing *any single* item from the latter.

Surveys of these notions, as well as other fairness notions for allocating indivisible goods, can be found, e.g., in (Amanatidis et al., 2023).

**Envy-free allocation via queries.** In most of the existing literature on envy-freeness (and more generally, fair division), it is assumed that the entire set of valuations is known in advance. However, a recent line of works has sought to understand various notions of *query complexity*. In particular, under additive valuations, *noise-free queries on bundles of items* were studied in (Oh et al., 2021; Bu et al., 2024), with (Oh et al., 2021) considering value-based queries and (Bu et al., 2024) considering comparison-based queries. In both cases, EF1 was shown to be achievable using a logarithmic number of queries. For the stronger notion of EFX, the query complexity increases to linear under additive valuations (Oh et al., 2021) and exponential under more general (non-additive) valuations (Plaut & Roughgarden, 2020a).

The preceding works are all fundamentally different from ours due to the queries being *noiseless* and being applied to *bundles of items* rather than individual items. Somewhat closer to our work is a recent study of the round-robin algorithm under additive valuations with potentially noisy queries (Li et al., 2025). These authors consider comparison-based queries to pairs of items, as well as value-based queries to individual items, giving various upper and lower bounds with at least an $nm$ dependence on the number of agents $n$ and items $m$. The major differences compared to our work are outlined as follows:

- They focus on *exactly implementing the round-robin algorithm*, which is one specific algorithm for attaining the EF1 guarantee. In contrast, we are interested in the conditions under which there exists an algorithm (not necessarily related to round-robin) that can achieve EF.

- Their query models are completely distinct from ours. In the comparison-based model, they assume that a single bit is received indicating which of two items is preferred by a given agent, possibly flipped by noise with some constant probability. In the value-based model, they crucially assume that the *exact valuation is observed with constant probability*. Thus, in both cases, repeated queries and a majority vote suffice to get the *exact correct answer*. In contrast, we study an *additive Gaussian noise* model, in which no matter how many queries we perform and collate, we will never know the exact valuations. This distinction is fundamental and turns out to be crucial.

- To make our (more challenging) problem feasible, we introduce a "gap" parameter indicating how negative the (unknown) optimal envy is, i.e., how far the corresponding allocation is from violating EF. Under the assumption of such a gap, we construct algorithms that *do not need to query every item*. In contrast, the lower bounds in (Li et al., 2025) reveal that, as one would expect, implementing the round-robin algorithm exactly requires querying every item at least once.

We note that the above-mentioned comparison-based query model is an instance of *dueling bandit feedback* (Sui et al., 2018), and the additive Gaussian noise model is an instance of *regular multi-armed bandit feedback* (Lattimore & Szepesvári, 2020, Ch. 4). We proceed to survey some other (less closely related) works that adopt a multi-armed bandit viewpoint.

**Other bandit-based settings involving envy-freeness.** In (Procaccia et al., 2024), a setting was considered in which items arrive sequentially and are allocated to agents in an online manner. Their goal involves achieving envy-freeness (or proportionality) *in expectation*, but the main objective

---

tional disputes involve two agents. See (Plaut & Roughgarden, 2020b, Sec. 1.1.1) for some discussion on the importance of this setting. The assumption of additive utilities is also very common in the literature.

[3]We use $\widetilde{O}(\cdot)$, $\widetilde{\Omega}(\cdot)$, and $\widetilde{\Theta}(\cdot)$ for asymptotic notation that hides logarithmic factors.

itself is a cumulative social welfare measure. Note that in our setup, envy-freeness in expectation could be obtained trivially (with no queries) by simply assigning items uniformly at random. Other differences in their work include only having finitely many "types" of item, and having to allocate items immediately as they arrive (i.e., the online setting). Other related works in the online setting, typically seeking objectives based on Nash welfare, include (Sinha et al., 2023; Bhattacharya et al., 2024; Yamada et al., 2024; Schiffer & Zhang, 2025; Verma et al., 2026).

In (Peters et al., 2022), a robust rent division problem is studied, with the goal of obtaining envy-free allocations robust to misspecified values. A major difference from our work is that they specifically focus on a "room allocation" problem in which every agent gets one room (i.e., a "matching" is formed). In contrast, we focus on a two-agent problem with arbitrarily many items. At a technical level, their query complexity indicates taking $O\left(\frac{1}{\varepsilon^2}\right)$ samples of each (agent, room) pair to attain a probability of envy-freeness within $\varepsilon$ of optimal, whereas in our problem we can often have fewer queries than items.

**Repeated allocation problems with bandit feedback.** Another line of work on item allocation with bandit feedback has considered scenarios where every round consists of proposing an *entire allocation*, rather than querying just one item and/or agent. The goal is typically to optimize some long-term measure of fairness. For instance, see (Talebi & Proutiere, 2018) for a study of proportional fairness in allocating tasks to servers, (Lim et al., 2024) for an egalitarian matching-based assignment problem, and (Harada et al., 2025) for a related problem of maximizing the minimum utility aggregated across a long sequence of allocations.

**Other fairness notions in bandit algorithms.** For other issues of fairness in bandit algorithms (not involving item allocation), we refer the interested reader to (Li et al., 2019; Hossain et al., 2021; Patil et al., 2021; Banihashem et al., 2023; Barman et al., 2023; Sawarni et al., 2023; Russo & Vannella, 2024) and the references therein. To name just one example, in (Hossain et al., 2021) each arm is valued differently by different agents, and the goal is to identify a distribution over the arms that maximizes the Nash welfare. Since item allocation is central to our work but is not considered in these works, we do not delve into the details.

### 1.2. Our Contributions

We formulate the problem of finding envy-free allocations from noisy valuation queries, focusing on the two-agent setting with Gaussian noise and valuations in $[0, 1]$. With $m$ denoting the number of items and $\Delta$ denoting the (unknown) optimal negative envy, our results are outlined as follows:

- In Section 3, we start with a "naive" analysis based on item-by-item confidence intervals, and show that a conceptually simple algorithm succeeds with $q = O\left(\frac{m^3}{\Delta^2}\right)$ queries. This is not one of our main contributions, but rather serves to highlight the suboptimality of a naive approach compared to our main algorithm.

- In Section 4, we provide our main algorithmic upper bound of $q = O\left(\frac{m^{2.5}}{\Delta^2}\right)$ queries whenever $\Delta \gg m^{1/4} \log^2 m$ (a condition that we will discuss in Remark 2 therein). While we maintain simplicity in the querying strategy (a uniform allocation, with suitable item subsampling if $q < m$) and the allocation rule (item-by-item thresholding), the mathematical analysis is significantly more challenging. Components of the analysis include bounding the assignment probabilities, carefully choosing the allocation strategy's threshold to "balance" the two kinds of envy, quantifying the difference between the true and estimated valuations, and (in the case that $q < m$) bounding the effect of the above-mentioned subsampling. See Section 4 for a more detailed overview.

- In Section 5, we present our algorithm-independent lower bound of $\widetilde{\Omega}\left(\frac{m^{2.5}}{\Delta^2}\right)$. This is broadly based on tools from multiple hypothesis testing, but with several unique aspects to capture the fact that attaining envy-freeness does not necessarily require estimating all (or even most) items accurately. One central idea is to have items that are *very slightly* favored by one agent and disfavored by the other, which necessitates giving sufficiently many of these items to the agent who prefers them. However, this idea alone fails to give a tight result, and we address this by also including *some items more strongly favored/disfavored by both agents*. It turns out that this creates "random fluctuations" in the envy that will need to be outweighed by the allocation of the other items. See Section 5 for a more detailed overview.

Collectively, these results establish that

> *the correct scaling on the optimal number of queries is $\widetilde{\Theta}\left(\frac{m^{2.5}}{\Delta^2}\right)$,*

at least when $\Delta \gg m^{1/4} \log^2 m$. Moreover, the upper bound is based on a non-adaptive querying strategy and a polynomial-time allocation strategy, whereas the lower bound holds even under adaptive queries and arbitrary computational complexity. We note that the seemingly unconventional $\frac{m^{2.5}}{\Delta^2}$ scaling can be more naturally viewed as $\frac{\sqrt{m}}{(\Delta/m)^2}$ with $\Delta/m$ representing a "normalized gap"; indeed, quadratic dependencies on gaps are ubiquitous in pure exploration problems for multi-armed bandits (Lattimore &

Szepesvári, 2020). Some intuition on the $\sqrt{m}$ numerator will be given in Section 4.

While the above discussion pertains to a constant noise level, we will also cover the case of general noise levels in Section 6. Our results can be translated to the fairness notion of proportionality; we discuss this in Section 7.

## 2. Problem Setup

We consider fairly allocating $m$ indivisible items between two agents $a$ and $b$, where the agents have their respective utility $u_i^a$ and $u_i^b$ for each item $i$, and $u_i^a, u_i^b \in [0, 1]$. Here the restriction to $[0, 1]$ is for convenience, and could be obtained from any interval of the form $[0, u_{\max}]$ by rescaling. Our problem setup could naturally be extended to more than two agents, but the two-agent setting is a fundamental starting point that already comes with considerable technical challenges. See Section 7 for some further discussion on the $n$-agent scenario.

We consider *additive utilities*, meaning that for $\nu \in \{a, b\}$, the overall value that agent $\nu$ assigns to a set of items $S$ is $\sum_{i \in S} u_i^\nu$. We define an *allocation* of the $m$ items as any partition $\mathcal{A} = (\mathcal{A}_a, \mathcal{A}_b)$ with $\mathcal{A}_a \cup \mathcal{A}_b = [m]$ and $\mathcal{A}_a \cap \mathcal{A}_b = \emptyset$, which means $\mathcal{A}_a$ is allocated to Agent $a$ and $\mathcal{A}_b$ to Agent $b$. For any such allocation, the *envy* from Agent $a$ to Agent $b$ is defined as

$$\text{Envy}_{a \to b}(\mathcal{A}) = \sum_{i \in \mathcal{A}_b} u_i^a - \sum_{i \in \mathcal{A}_a} u_i^a,$$

and Agent $a$ envies Agent $b$ if $\text{Envy}_{a \to b}(\mathcal{A}) > 0$ (and similarly with the roles of Agents $a$ and $b$ reversed.) The overall envy of the allocation $\mathcal{A}$ is defined as

$$\text{Envy}(\mathcal{A}) = \max\{\text{Envy}_{a \to b}(\mathcal{A}), \text{Envy}_{b \to a}(\mathcal{A})\}.$$

When $\text{Envy}(\mathcal{A}) \le 0$, neither agent envies the other, and we call $\mathcal{A}$ an *envy-free allocation*.

While envy-freeness, as well as variants such as "envy-freeness up to one item", has been studied extensively in settings with perfectly known valuations, the presence of noise turns out to make such a goal highly challenging in general, unless a very large number of queries is taken. (Our lower bounds will formalize this claim.) Accordingly, in order to avoid overly pessimistic "worst-case" thinking, we assume that the (unknown) optimal allocation has *strictly negative* envy with some gap $\Delta > 0$:

$$\text{OptEnvy} = \min_{\mathcal{A}} \text{Envy}(\mathcal{A}) \le -\Delta. \qquad (1)$$

Since finding the precise optimal allocation may be prohibitive, we set the more modest goal of *finding any envy-free allocation*, hence why we refer to $\Delta$ as a "gap". That is, our goal is to obtain an allocation $\widehat{\mathcal{A}}$ satisfying $\text{Envy}(\widehat{\mathcal{A}}) \le 0$ based on the noisy queries.

The algorithm performs some number of queries $q$ indexed by $1, \ldots, q$. Specifically, at each iteration, we allow the algorithm to query an item $i$ and obtain a pair of noisy observations.[4] When a given item is sampled for the $t$-th time, we denote its outcome by $y_t = (y_{i,t}^a, y_{i,t}^b)$. We consider an additive Gaussian noise model, in which

$$y_{i,t}^\nu \sim N(u_i^\nu, \sigma^2) \quad \text{for} \quad \nu \in \{a, b\}, \qquad (2)$$

where $\sigma^2 > 0$ is the noise variance. We assume that all query outcomes are independent of one another (including independence of $y_{i,t}^a$ from $y_{i,t}^b$). We seek to minimize the number of noisy queries required to achieve an envy-free allocation with high probability; the total number of queries is denoted by $q$. We use log to denote the natural logarithm.

**Remark 1.** *(Discussion on noise model)* The presence of noise is motivated by scenarios in which we have imperfect information regarding the user valuations, e.g., due to users' perceived valuations being influenced by external factors, or due to precise valuations being too costly to obtain. Naturally, the ideal mathematical model may vary vastly depending on the precise application. Our particular noise model has two main notable properties that deserve discussion: (i) independence between queries, and (ii) being additive Gaussian.

Regarding property (i), the independence assumption has a very strong precedent from a theoretical perspective, as it has been adopted in the overwhelming majority of works in noisy allocation, multi-armed bandits, and so on (surveyed in Section 1.1). However, it is important to keep in mind that the assumption is not necessarily true in practice; for example, if we query the same user multiple times, their next result reported may very well be influenced by their past reportings. On the other hand, allowing arbitrary dependencies may considerably complicate the analysis, as well as diminish the benefit of repeating the same query multiple times. Notably, some of our results for "low query budget" regimes will be based on only querying any given item at most once (e.g., see Appendix C.4), in which case there are no repeated queries.

Regarding property (ii), additive Gaussian noise is undoubtedly one of the most fundamental and ubiquitous noise models in diverse statistical problems such as multi-armed bandits, statistical estimation, error-correcting codes for communication, and so on. The analysis in our main upper bound does use the specific Gaussianity property, with generalizations such as sub-Gaussian being conceivable but non-trivial. Since the analysis is already highly challenging, we leave such generalizations to future work. Note also that for the *lower bound*, adopting a specific widely-used noise

---

[4]An alternative setup would be that in which the algorithm only queries a single (agent, item) pair. The query complexities of the two settings trivially match to within a factor of 2.

---

**Algorithm 1** Repeated Sampling for Envy-Free Allocation

---

1: **Input:** Number of queries $q$
2: Query each item $\tau = q/m$ times and observe $\{y_{i,t}^a, y_{i,t}^b\}_{t=1}^{\tau}$.
3: Compute valuation estimates $v_i^{\nu} = \frac{1}{\tau} \sum_{t=1}^{\tau} y_{i,t}^{\nu}$ for each $\nu \in \{a, b\}$ and $i \in [m]$.
4: Loop over all possible allocations $\mathcal{A} = (\mathcal{A}_a, \mathcal{A}_b)$ and return the one with the highest value of $\min\{v_{\mathcal{A}_a}^a - v_{\mathcal{A}_b}^a, v_{\mathcal{A}_b}^b - v_{\mathcal{A}_a}^b\}$, where $v_S^{\nu} = \sum_{i \in S} v_i^{\nu}$ for $\nu \in \{a, b\}$ and $S \in \{\mathcal{A}_a, \mathcal{A}_b\}$.

---

model (rather than the hardest distribution within a more general class) turns into a strength rather than a limitation.

## 3. An Initial Suboptimal Upper Bound

We first consider an algorithm (presented in Algorithm 1) based on repeated sampling and straightforward confidence intervals on the item utilities. With $q$ queries and $m$ items, we query each item $\tau = q/m$ times, and average the observations $(y_{i,t}^a, y_{i,t}^b)$ across $t = 1, \ldots, \tau$ to form estimated utilities $(v_i^a, v_i^b)$. Then, we compute the estimated envy for each possible allocation and return the one with the lowest estimated envy (i.e., highest estimated negative envy).

The following theorem states a sufficient number of queries to ensure the success of this algorithm. (Recall that we use $\widetilde{O}(\cdot)$, $\widetilde{\Omega}(\cdot)$, and $\widetilde{\Theta}(\cdot)$ for asymptotic notation that hides logarithmic factors.)

**Theorem 1.** *For any $\delta \in (0, 1)$, Algorithm 1 outputs an envy-free allocation with probability at least $1 - \delta$ when the number of queries is set to*

$$q = m\lceil 32\sigma^2 \log(4m/\delta) \cdot m^2/\Delta^2 \rceil = \widetilde{O}\left(\frac{m^3}{\Delta^2}\right). \quad (3)$$

The proof is given in Appendix B, and is based on a simple analysis that forms a confidence interval on the valuation of each item, and then computes a confidence width for each *bundle* that equals the sum of individual confidence widths. Based on these confidence intervals on bundles, we can conclude that the decision rule (output the bundle with the highest estimated negative envy) will be envy-free with high probability when enough queries are taken.

While Theorem 1 is a useful starting point, it has two major limitations:

- The $m^3$ dependence turns out to be suboptimal; we will see that the correct dependence is $m^{2.5}$.

- The algorithm involves a brute force search over all possible allocations $\mathcal{A}$, and thus (at least in its current form) it is not computationally efficient.

Regarding the first dot point, a key weakness in the naive approach is the reliance on each individual item's confidence interval when estimating the value of an entire bundle, and implicitly assuming that the errors always accumulate in the worst manner possible (thus multiplying the amount of error by the bundle size). Note that this weakness concerns the *analysis* rather than the algorithm itself, so it is conceivable that Algorithm 1 could satisfy stronger guarantees if analyzed more carefully.

Our main upper bound will overcome these limitations via a more carefully-designed algorithm with a more careful (and significantly more challenging) mathematical analysis.

## 4. An Improved Upper Bound

We now state our main upper bound showing that the $\frac{m^3}{\Delta^2}$ dependence can be reduced to $\frac{m^{2.5}}{\Delta^2}$ (at least when $\Delta$ is not too small), and achieving this with polynomial running time. We state the result for a constant noise level here, but also provide a generalization to the regimes $\sigma^2 = o(1)$ and $\sigma^2 = \omega(1)$ in the proof (see Section 6 for a detailed discussion and comparison).

**Theorem 2.** *For any constant noise level $\sigma^2 > 0$, when $\Delta \geq m^{1/4} \log^2 m$ and $\Delta \leq Cm$ for sufficiently small $C$, there exists a polynomial-time algorithm that outputs an envy-free allocation with probability $1 - o(1)$ as $m \to \infty$ while using a number of queries at most $q \leq \widetilde{O}\left(\frac{m^{2.5}}{\Delta^2}\right)$. Moreover, these queries can be taken non-adaptively.*

We proceed to outline the algorithm and analysis, deferring the details to Appendix C. (We also discuss the assumption $\Delta \geq m^{1/4} \log^2 m$ in Remark 2 below.) We will split up the proof according to whether or not there are enough queries to sample every item once; by the scaling $q = \widetilde{O}\left(\frac{m^{2.5}}{\Delta^2}\right)$, this amounts to having $\Delta \lesssim m^{3/4}$ vs. $\Delta \gtrsim m^{3/4}$, to within logarithmic factors. We refer to these as the regimes of "smaller $\Delta$" and "larger $\Delta$" respectively.

In the smaller $\Delta$ regime, we adopt the same initial steps as Algorithm 1: Sample every item $\tau = \frac{q}{m}$ times, and compute its valuation estimates $v_i^{\nu} = \frac{1}{\tau} \sum_{t=1}^{\tau} y_{i,t}^{\nu}$ for each $\nu \in \{a, b\}$. However, we do not use these to estimate the total valuations of bundles, but instead, we allocate via simple thresholding on an item-by-item basis:

$$\text{Assign item } i \text{ to Agent} \begin{cases} a & \text{if } cv_i^a - v_i^b > 0 \\ b & \text{otherwise,} \end{cases} \quad (4)$$

for some parameter $c > 0$. Naturally, this means that items with higher $v_i^{\nu}$ are favored for agent $\nu \in \{a, b\}$, and the parameter $c > 0$ controls how much we prioritize one agent vs. the other. One might be tempted to set $c = 1$ to treat both agents equally, but this is too naive unless the valuations have some suitable "symmetry"; for instance, it may fail

if one agent has uniformly higher valuations than the other agent for all items.

The analysis itself is rather technical, so we only outline some of the main ideas:

- We observe that $cv_i^a - v_i^b$ is Gaussian due to the Gaussian noise, and using this, we can precisely characterize the assignment probabilities of a given item. By doing so and summing over the items, we show that with high probability, there is a certain amount of "total negative envy with respect to the true valuations" (depending on $c$) summed over both $a \to b$ and $b \to a$ with suitable weighting.

- We establish that this negative envy can be made "balanced" (for suitably-chosen $c$) with respect to the estimated valuations, in the sense of making the $a \to b$ envy and $b \to a$ envy be individually low as opposed to just their combination. The rough idea is that these are imbalanced towards one agent for small $c$, the other agent for large $c$, and the behavior as $c$ varies can be shown to be "sufficiently smooth" to ensure the right balance somewhere in between. We also show that the required choice of $c$ only depends on the estimated valuations, meaning it can be computed by the algorithm.

- We use concentration arguments to relate the estimated valuations to the true ones, and use this finding to characterize how large the number of queries $q$ should be to maintain sufficient balancedness with respect to the *true* valuations.

Next, we discuss the larger $\Delta$ regime in which there are not enough queries to sample every item once. In this case, the idea is to sample a random subset of $q < m$ items once each and obtain the guarantee from the "every item gets sampled" regime restricted to those items, thus with the number of items $m$ replaced by $q$, and with $\Delta$ replaced by roughly $\Delta \frac{q}{m}$ (which is formalized using a concentration argument). The items that are not sampled are allocated to each agent independently with probability $\frac{1}{2}$ each, i.e., completely randomly. Such an allocation is "fair on average" but has fluctuations of size roughly $\sqrt{m}$ by a central limit theorem argument. We then require $q$ to be large enough such that the "negative envy" gained from the sampled items outweighs these fluctuations.

**Remark 2.** *(Assumption on $\Delta$)* In general, $\Delta$ lies in the range $[0, m]$, meaning that the restriction to $\Delta \geq m^{1/4} \log^2 m$ captures "most" of the possible scaling regimes. Nevertheless, it would be of interest to handle $\Delta \leq O(m^{1/4})$ as well. The lower bound on $\Delta$ arises in our analysis for somewhat technical reasons: we want to control a quantity (related to "smoothness" in the second dot point above) to be at most $\widetilde{O}\left(\frac{m}{\sqrt{q}}\right)$, which we are only

able to achieve if $q \ll m^2$. In contrast, attaining the desired bound $q = \widetilde{O}\left(\frac{m^{2.5}}{\Delta^2}\right)$ when $\Delta \ll m^{1/4}$ requires us to have $q \gg m^2$. We also note that certain regimes of *sufficiently small* $\Delta$ may have computational barriers in view of the fact finding an envy-free allocation (i.e., $\Delta = 0$) is NP-hard due to a reduction from PARTITION (Lipton et al., 2004).

Overall, while we do not wish to confidently claim anything around this discussion, we expect that precluding the regime $\Delta \leq O(m^{1/4})$ is not merely an artifact/weakness in our analysis, but rather, that this regime requires fundamentally different algorithms (e.g., based on actually forming bundles rather than using the item-by-item approach of (4)).

## 5. Algorithm-Independent Lower Bound

In this section, we study algorithm-independent lower bounds on the number of queries necessary to achieve envy-freeness. Our main result of this section is Theorem 3 below stating an $\Omega\left(\frac{m^{2.5}}{\Delta^2}\right)$ lower bound, thus matching the upper bound to within logarithmic factors.

**Failure of naive approach.** Before outlining our techniques and stating the result formally, we give some motivating discussion. Motivated by lower bounds for multi-armed bandits, a natural approach would be to have half the items be slightly favored by Agent $a$ and the other half slightly favored by Agent $b$, e.g., with utilities $\frac{1}{2} \pm \varepsilon$ for some small $\varepsilon > 0$. This leads to an optimal negative envy of $\Delta = m\varepsilon$, meaning $\varepsilon = \frac{\Delta}{m}$, and it is intuitively difficult to learn each item's "type" because $\varepsilon$ is small.

However, such an approach turns out to be insufficient to obtain the desired $\frac{m^{2.5}}{\Delta^2}$ dependence in the lower bound. The limitation is most easily seen when $\Delta \gg m^{3/4}$ (e.g., $\Delta = m^{0.9}$), in which case the desired bound satisfies $q \ll m$ and most items cannot be queried. By allocating the non-queried items uniformly at random, these items will amount to an *average* envy from $a$ to $b$ of zero, and similarly for the envy from $b$ to $a$. Moreover, a standard central limit theorem argument reveals that the deviations from zero are on the order of $\Theta(\varepsilon\sqrt{m})$ with high probability. As a result, the queried items must be allocated sufficiently well to overcome these fluctuations. As we further discuss below, a more carefully-designed hard instance can increase such fluctuations to $\Theta(\sqrt{m})$, thus indicating that $\Theta(\varepsilon\sqrt{m})$ is significantly smaller than ideal (since $\varepsilon \ll 1$ except when $\Delta = \Theta(m)$).

We note (without proof) that the above "naive" approach turns out to give a lower bound with a leading term of $\frac{m}{\Delta^{1.5}}$ for $\Delta \gg \sqrt{m}$, and $\frac{m^2}{\Delta^2}$ for $\Delta \ll \sqrt{m}$, both of which are worse lower bounds than $\frac{m^{2.5}}{\Delta^2}$.

**Our approach.** The idea of our refined lower bound construction is to have other types of items that considerably in-

crease the magnitude of the fluctuations (e.g., in the preceding example, to $\Theta(\sqrt{m})$ instead of only $\Theta(\varepsilon\sqrt{m})$). While less immediately obvious, the inclusion of such items will help even in the regime $\Delta \ll m^{3/4}$ in which $q$ is large enough to query every item. Specifically, we introduce *items that are (relatively strongly) favored/disfavored by both agents*, i.e., their utility is $\frac{1}{2} + \gamma$ for both or $\frac{1}{2} - \gamma$ for both, where typically $\gamma \gg \varepsilon$. We will keep the parameter $\gamma$ general throughout the analysis, but will end up choosing $\gamma = \frac{1}{2}$ when $\Delta \gg m^{3/4}$, and $\gamma = \Theta(\varepsilon m^{1/4})$ when $\Delta \ll m^{3/4}$.

In more detail, we design a randomized hard instance as in Table 1, containing four types of items. The items of the same type have the same utilities and the type of each item depends on (i) whether its index $i$ satisfies $i \leq \frac{m}{2}$, and (ii) a latent variable $X_i$ independently sampled from Bernoulli$(\frac{1}{2})$. The utilities involve parameters $\varepsilon, \gamma \in (0, 0.5]$, which possibly depend on $\Delta$ and $m$, and their exact values will be specified later.

|        | $i \leq m/2$ | | $i > m/2$ | |
|--------|-------------|-----------|-----------|-----------|
|        | $X_i = 1$ | $X_i = 0$ | $X_i = 1$ | $X_i = 0$ |
| $u_i^a$ | $\frac{1}{2} + \varepsilon$ | $\frac{1}{2} - \varepsilon$ | $\frac{1}{2} + \gamma$ | $\frac{1}{2} - \gamma$ |
| $u_i^b$ | $\frac{1}{2} - \varepsilon$ | $\frac{1}{2} + \varepsilon$ | $\frac{1}{2} + \gamma$ | $\frac{1}{2} - \gamma$ |

*Table 1.* Hard instance for the algorithm-independent lower bound. $X_i$ is independently sampled from Bernoulli$(\frac{1}{2})$; $\varepsilon$ and $\gamma$ are positive parameters that possibly depend on $\Delta$ and $m$.

The analysis is given in Appendix D, and leads to the following theorem for any constant noise level $\sigma^2 > 0$. A more general statement depending on $\sigma$ is given in Theorem 7 in Appendix D, and we discuss this $\sigma$ dependence in detail in Section 6.

**Theorem 3.** *Let $\sigma^2 > 0$ be fixed (not depending on $m$), and let $\Delta$ take any value in $(1, m/2)$.[5] Then, under the above randomized instance with suitably-chosen $\varepsilon$ and $\gamma$, for any (possibly adaptive and/or randomized) algorithm whose number of queries satisfies $q \leq O\left(\frac{m^{2.5}}{\Delta^2}\right)$ with a small enough implied constant, it holds with probability at least $1/3$ that (i) OptEnvy $\leq -\Delta$, and (ii) the algorithm's output has positive envy.*

## 6. General Noise Levels

In Theorems 2 and 3, we considered a fixed constant noise level $\sigma^2 > 0$, in particular satisfying $\sigma^2 = \Theta(1)$ as $m \to \infty$. In this section, we state and discuss further results (proved in the appendices) for general values of $\sigma$, possibly scaling as $o(1)$ or $\omega(1)$ with respect to $m$.

**Upper bound.** In the proof of Theorem 2, we will split the upper bound into two theorems depending on whether

---

[5]The lower bound of 1 is already very mild, but can be replaced by any fixed positive constant.

the number of queries $q$ is above or below the number of items $m$:

- The case $q \geq m$ is handled in Theorem 5 in Appendix C.3, which states that if $\Delta \geq m^{1/4} \log^2 m$, then a sufficient number of queries is

$$q = m\left\lceil \sigma^2\left(15\frac{m^{3/2}}{\Delta^2}\log m + \log^2 m\right)\right\rceil. \quad (5)$$

- The case $q < m$ is handled in Theorem 6 in Appendix C.4, which states that if (i) $\Delta^2 > 160\sigma m^{3/2} \log^2 m$, (ii) $\Delta^4 > 160^2 m^3 \sigma^4 \log^2 m$, and (iii) $\Delta < \min\{2\sigma^2 m, \sqrt{2}\sigma m\}$, then a sufficient number of queries is

$$q = \left\lceil \max\left\{160^2 \frac{m^4}{\Delta^4}\sigma^4 \log^2 m, 160\frac{\sigma m^{5/2}}{\Delta^2}\log^2 m\right\}\right\rceil. \quad (6)$$

**Lower bound.** A more general version of Theorem 3, stated as Theorem 7 in Appendix D, includes the dependence on the noise level, which we summarize here for convenience: We have a constant probability of failure whenever

$$q \leq \begin{cases} O\left(\frac{\sigma m^{2.5}}{\Delta^2}\right) & \text{when } \Delta = \omega(m^{3/4}) \\ O\left(\frac{\sigma^2 m^{2.5}}{\Delta^2}\right) & \text{when } \Delta = O(m^{3/4}) \end{cases} \quad (7)$$

with a sufficiently small implied constant. While this result is clearly loose for extremely small $\sigma$ (in particular becoming $q = 0$ when $\sigma = 0$), it turns out to be tight in broad scaling regimes, as we discuss below.

**Comparison.** Our upper bounds contain more conditions and $\max\{\cdot, \cdot\}$ terms than our lower bounds, and accordingly, the two do not always coincide. Nevertheless, we observe tightness in broad cases of interest, including the following:

- The first term in the parentheses in (5) is order-wise no smaller than the second whenever $\Delta = O\left(\frac{m^{3/4}}{\sqrt{\log m}}\right)$. Under this condition, we find that we are in the second case in (7) (i.e., $\Delta = O(m^{3/4})$), and we have matching upper and lower bounds to within an $O(\log m)$ factor.

- When the maximum in (6) is achieved by the second term (i.e., when $\Delta > \sqrt{160}m^{3/4}\sigma^{3/2}$), and when we are in the first case in (7) (i.e., when $\Delta = \omega(m^{3/4})$), we have matching upper and lower bounds to within an $O((\log m)^2)$ factor.

While this establishes broad scaling regimes in which our bounds are tight, they are not exhaustive. In the second dot point, we could have $\Delta = \omega(m^{3/4})$ and yet $\Delta < \sqrt{160}m^{3/4}\sigma^{3/2}$ due to $\sigma = \omega(1)$ being very large. Moreover, perhaps more fundamentally, our upper bounds are not applicable when $\Delta \leq O(m^{1/4})$, and we discussed in Remark 2 how this may be a fundamental limitation of our choice of algorithm.

# 7. Conclusion

We have introduced the problem of envy-free allocation with noisy queries, and established upper and lower bounds on the sample complexity (for the two-agent setting with additive utilities and Gaussian noise) in terms of the number of items $m$ and the optimal negative-envy $\Delta$. In particular, we established that the optimal number of queries is $\widetilde{\Theta}\big(\frac{m^{2.5}}{\Delta^2}\big)$ when $\Delta \gg m^{1/4}$. We believe that our work opens up several directions for further research, including (i) fully understanding the case $\Delta \ll m^{1/4}$ (see Remark 2 for discussion), (ii) extending to more than two agents and/or more general noise models, (iii) extending beyond additive valuations, and (iv) extending to other fairness notions.

Regarding the $n$-agent scenario for $n > 2$, a notable challenge in the upper bound is that the allocation rule (4) amounts to checking whether the ratio of estimated valuations exceeds a given threshold, but with $n$ agents there are $\binom{n}{2}$ relevant ratios, and it is unclear how to combine them, or even whether these are the right quantities to work with (e.g., it is conceivable that "beyond pairwise" information is also needed). For the lower bound, one could try to generalize the hard instance from Table 1, e.g., to contain $n$ pairs of item types each favored/disfavored by only one of the agents, and a further pair favored/disfavored by all agents. However, significant effort is still likely to be needed in adapting the analysis and precisely determining which hard instance gives a tight result.

Finally, regarding other fairness notions, we comment on another well-known fairness notion, *proportionality* (e.g., see (Amanatidis et al., 2023)). An allocation is said to be *proportional* if it gives each agent a utility least as high as the agent's *proportional share*, defined as $1/n$ times their utility for the set of all items. In the case of $n = 2$ agents and additive utilities, envy-freeness and proportionality are equivalent: A direct comparison of the fairness definitions yields that an agent has an envy of $r > 0$ (resp., $r < 0$) if and only if the agent's utility is below (resp., above) their proportional share by $r/2$. Therefore, all of our results can be directly transferred to proportionality. As with envy-freeness, extending these results to $n$ agents for proportionality is an interesting direction for future work.

# Acknowledgment

Y.L. is supported by NTU Start-up Grant #025311-00001. J.S. is supported by the Singapore National Research Foundation (NRF) under its AI Visiting Professorship programme. W.S. is supported by the Singapore Ministry of Education under grant number MOE-T2EP20221-0001 and by an NUS Start-up Grant.

# Impact Statement

This paper presents work whose goal is to advance the field of Machine Learning. There are many potential societal consequences of our work, none which we feel must be specifically highlighted here.

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

# Appendix

## A. Probabilistic Tools

Here we state some useful probabilistic tools that are used in our analysis.

**Lemma 1.** (Chernoff bound for Gaussian RVs (Vershynin, 2018, Ch. 2)) *Let $y_1, \ldots, y_T$ be independent random variables such that $y_t \sim N(\mu, \sigma^2)$. Then, it holds for any $\varepsilon > 0$ that*

$$\mathbb{P}\Big[\Big|\frac{1}{T}\sum_{t=1}^{T} y_t - \mu\Big| \geq \varepsilon\Big] \leq 2\exp\Big(-\frac{T\varepsilon^2}{2\sigma^2}\Big). \tag{8}$$

**Lemma 2.** (Bernstein's inequality for bounded RVs (Vershynin, 2018, Ch. 2)) *Let $y_1, \ldots, y_T$ be independent and identically distributed zero-mean random variables satisfying $|y_i| \leq b$ almost surely, and $\mathrm{Var}[y_i] = \sigma_y^2$. Then, we have*

$$\mathbb{P}\Big[\Big|\frac{1}{T}\sum_{t=1}^{T} y_t\Big| \geq \varepsilon\Big] \leq 2\exp\Big(-\frac{T\varepsilon^2/2}{\sigma_y^2 + b\varepsilon/3}\Big). \tag{9}$$

**Lemma 3.** (Berry–Esseen theorem for non-identical RVs (Petrov, 1995, Thm. 5.6)) *Let $y_1, \ldots, y_T$ be independent random variables with each $y_t$ having mean $\mu_t$, variance $\sigma_t^2$, and finite third absolute moment. Define*

$$V_T = \sum_{t=1}^{T} \sigma_t^2, \quad \Psi_T = \sum_{t=1}^{T} \mathbb{E}[|y_t - \mu_t|^3]. \tag{10}$$

*Then, the shifted normalized summation $Z_T = \frac{\sum_{t=1}^{T}(y_t - \mu_t)}{\sqrt{V_T}}$ satisfies*

$$\sup_{\gamma \in \mathbb{R}} \big|\mathbb{P}[Z_T \leq \gamma] - \mathbb{P}[N(0,1) \leq \gamma]\big| \leq \frac{C\Psi_T}{V_T^{3/2}} \tag{11}$$

*for some absolute constant $C$. In particular, if $\Psi_T = O(T)$ and $V_T = \Omega(T)$, then the right-hand side is $O\big(\frac{1}{\sqrt{T}}\big)$.*

**Lemma 4.** (Assouad's lemma (Duchi, 2023, Sec. 9.5)) *Let $\Theta = \{0,1\}^d$ for some integer $d \geq 1$, let $\{P_\theta : \theta \in \Theta\}$ be a family of probability measures taking values on some space $\mathcal{Y}$, and let $\hat{\theta} : \mathcal{Y} \to \Theta$ be any estimator of $\Theta$ based on an outcome in $\mathcal{Y}$. Define the Hamming distance $d_H(\theta, \theta') = \sum_{i=1}^{d} \mathbf{1}\{\theta_i \neq \theta_i'\}$, and the optimal expected risk under a uniform prior as[6]*

$$\overline{M}(\Theta) = \inf_{\hat{\theta}} \frac{1}{|\Theta|} \sum_{\theta \in \Theta} \mathbb{E}_\theta\big[d_H\big(\theta, \hat{\theta}(Y)\big)\big], \tag{12}$$

*where $\mathbb{E}_\theta[\cdot]$ signifies that $Y \sim P_\theta$, and the infimum is over all estimators $\hat{\theta}$. Then, we have*

$$\overline{M}(\Theta) \geq \frac{1}{2} \sum_{i=1}^{d} \Big(1 - \|P_i^{(0)} - P_i^{(1)}\|_{\mathrm{TV}}\Big), \tag{13}$$

*where $P_i^{(\kappa)} = \frac{1}{2^{d-1}} \sum_{\theta : \theta_i = \kappa} P_\theta$ for $\kappa \in \{0,1\}$. That is, $P_i^{(\kappa)}$ is the conditional probability measure on $\mathcal{Y}$ given $\theta_i = \kappa$ when $\theta$ is uniformly random. Moreover, $\|P - Q\|_{\mathrm{TV}}$ denotes the total variation distance between $P$ and $Q$.*

**Lemma 5.** (Convexity of KL divergence (Polyanskiy & Wu, 2025, Thm. 5.1)) *The KL divergence $D(P\|Q)$ is jointly convex in its arguments, thus implying the following via Jensen's inequality: Let $\{P_j\}$ and $\{Q_j\}$ be two countable[7] collections of distributions on a common alphabet, let $\lambda$ be an arbitrary distribution over the indices $\{j\}$ (i.e., the values $\lambda_j$ are non-negative and sum to 1), and define the mixture distributions $P(x) = \sum_j \lambda_j P_j(x)$ and $Q(x) = \sum_j \lambda_j Q_j(x)$. Then, we have*

$$D_{\mathrm{KL}}(P\|Q) \leq \sum_j \lambda_j D_{\mathrm{KL}}(P_j\|Q_j). \tag{14}$$

---

[6]In (Duchi, 2023, Thm. 9.5.2) the lemma is stated in terms of the minimax risk instead (i.e., with $\max_\theta$ in place of $\frac{1}{|\Theta|}\sum_{\theta \in \Theta}$), but the proof in (Duchi, 2023, Sec. 9.6.3) is based on lower bounding the minimax risk by the average risk.

[7]The counterpart with continuous indices also holds, but we will only need the discrete version, and it is more convenient to state in this form.

**Lemma 6.** (Consequence of chain rule for KL divergence, e.g., Ex. 15.8 and Eq. (38.22) in (Lattimore & Szepesvári, 2020)) *Fix positive integers $k$ and $n$. Consider any distributions $P_1, P_1', \ldots, P_k, P_k'$ on a common alphabet, and write $P = (P_1, \ldots, P_k)$ and $P' = (P_1', \ldots, P_k')$. Then, consider an arbitrary (possibly adaptive) algorithm that, at each time indexed by $t = 1, \ldots, n$, queries an index $i_t \in \{1, \ldots, k\}$ and observes a random sample $y_t$ from the $i_t$-th distribution (i.e., from $P_{i_t}$ under $P$, or from $P_{i_t}'$ under $P'$). Let $Y = (y_1, \ldots, y_n)$ be the resulting sequence of outcomes, and let $P_Y$ and $P_Y'$ be the corresponding joint distributions on these outcomes under $P$ and $P'$. Then, it holds that*

$$D_{\mathrm{KL}}(P_Y \| P_Y') = \sum_{i=1}^{k} \mathbb{E}_P[N_i] D_{\mathrm{KL}}(P_i \| P_i'), \tag{15}$$

*where $N_i$ is the (random) number of times the $i$-th distribution is queried. In particular, if $P$ and $P'$ only differ in a single distribution $P_i$ (i.e., $P_j = P_j'$ for all $j \neq i$), then this simplifies to*

$$D_{\mathrm{KL}}(P_Y \| P_Y') = \mathbb{E}_P[N_i] D_{\mathrm{KL}}(P_i \| P_i'). \tag{16}$$

## B. Proof of Theorem 1 (Basic Upper Bound)

Recall that Theorem 1 states that Algorithm 1 succeeds with probability at least $1-\delta$ when $q = m\lceil 32\sigma^2 \log(4m/\delta) \cdot m^2/\Delta^2 \rceil$. Since this algorithm consists of sampling each item repeatedly, we make use of a standard Chernoff-type bound for Gaussian random variables (see Lemma 1 in Appendix A). Specifically, for any $\delta \in (0,1)$, letting $\delta_0 = \frac{\delta}{2m}$, Lemma 1 implies that for any fixed $(\nu, i)$ pair, after performing $\tau = \frac{q}{m}$ queries, it holds that

$$\mathbb{P}\left[\left|v_i^\nu - u_i^\nu\right| \geq \sqrt{\frac{2\sigma^2 \log(2/\delta_0)}{\tau}}\right] \leq \delta_0, \tag{17}$$

where $v_i^\nu = \frac{1}{\tau}\sum_{t=1}^{\tau} y_{i,t}^\nu$, and $u_i^\nu$ is the true utility. By a union bound over $i = 1, \ldots, m$ and the triangle inequality, defining $\varepsilon_S = |S|\sqrt{\frac{2\sigma^2 \log(2/\delta_0)}{\tau}}$ and writing $v_S^\nu = \sum_{i \in S} v_i^\nu$ and $u_S^\nu = \sum_{i \in S} u_i^\nu$, we have

$$\mathbb{P}\left[\bigcup_{\nu \in \{a,b\}, S \subseteq [m]} \left(\left|v_S^\nu - u_S^\nu\right| \geq \varepsilon_S\right)\right] \leq \sum_{\nu \in \{a,b\}} \sum_{i=1}^{m} \mathbb{P}\left[\left|v_i^\nu - u_i^\nu\right| \geq \sqrt{\frac{2\sigma^2 \log(2/\delta_0)}{t}}\right] \tag{18}$$

$$\leq 2m\delta_0 \tag{19}$$

$$= \delta. \tag{20}$$

Hence, with probability at least $1 - \delta$, the following holds for all $\nu \in \{a,b\}$ and $S \subseteq [m]$:

$$|v_S^\nu - u_S^\nu| \leq \varepsilon_S. \tag{21}$$

We proceed conditioned on this being true.

Next, letting $\mathcal{A}$ be the allocation output by the algorithm, we have

$$-\mathrm{Envy}_{a \to b}(\mathcal{A}) = u_{\mathcal{A}_a}^a - u_{\mathcal{A}_b}^a \geq (v_{\mathcal{A}_a}^a - \varepsilon_{\mathcal{A}_a}) - (v_{\mathcal{A}_b}^a + \varepsilon_{\mathcal{A}_b}), \tag{22}$$

$$-\mathrm{Envy}_{b \to a}(\mathcal{A}) = u_{\mathcal{A}_b}^b - u_{\mathcal{A}_a}^b \geq (v_{\mathcal{A}_b}^b - \varepsilon_{\mathcal{A}_b}) - (v_{\mathcal{A}_a}^b + \varepsilon_{\mathcal{A}_a}), \tag{23}$$

where the inequality follows from (21). Combining these inequalities, we have

$$-\mathrm{Envy}(\mathcal{A}) = \min\{-\mathrm{Envy}_{a \to b}(\mathcal{A}), -\mathrm{Envy}_{b \to a}(\mathcal{A})\} \tag{24}$$

$$\geq \min\{v_{\mathcal{A}_a}^a - v_{\mathcal{A}_b}^a, v_{\mathcal{A}_b}^b - v_{\mathcal{A}_a}^b\} - (\varepsilon_{\mathcal{A}_a} + \varepsilon_{\mathcal{A}_b}) \tag{25}$$

$$\geq \min\{v_{\mathcal{A}_a^*}^a - v_{\mathcal{A}_b^*}^a, v_{\mathcal{A}_b^*}^b - v_{\mathcal{A}_a^*}^b\} - (\varepsilon_{\mathcal{A}_a} + \varepsilon_{\mathcal{A}_b}), \tag{26}$$

where the last step holds with $\mathcal{A}^*$ being the (unknown) optimal allocation, due to the fact that the algorithm chooses $\mathcal{A}$ to maximize $\min\{v_{\mathcal{A}_a}^a - v_{\mathcal{A}_b}^a, v_{\mathcal{A}_b}^b - v_{\mathcal{A}_a}^b\}$. By bounding the estimates in terms of the true values in the same way as (22) (but in the opposite direction), we can further weaken (26) to

$$-\mathrm{Envy}(\mathcal{A}) \geq \min\{u_{\mathcal{A}_a^*}^a - u_{\mathcal{A}_b^*}^a, u_{\mathcal{A}_b^*}^b - u_{\mathcal{A}_a^*}^b\} - (\varepsilon_{\mathcal{A}_a} + \varepsilon_{\mathcal{A}_b} + \varepsilon_{\mathcal{A}_a^*} + \varepsilon_{\mathcal{A}_b^*}) \tag{27}$$

$$\geq \Delta - 4\varepsilon_{\max}, \tag{28}$$

where we recall that the optimal allocation has negative envy at least $\Delta$, and we define $\varepsilon_{\max}$ as the largest possible $\varepsilon_S$ value, namely, the one that would correspond to a set of size $|S| = m$. Thus, we have established that the allocation is envy-free provided that $\varepsilon_{\max} \leq \frac{\Delta}{4}$.

Let $\varepsilon_0 = \sqrt{\frac{2\sigma^2 \log(2/\delta_0)}{\tau}}$ denote the approximation error for any single item after performing $\tau$ queries. To guarantee $\varepsilon_{\max} \leq \frac{\Delta}{4}$, it suffices to have $\varepsilon_0 \leq \frac{\Delta}{4m}$, i.e.,

$$\varepsilon_0 = \sqrt{\frac{2\sigma^2 \log(2/\delta_0)}{\tau}} \leq \frac{\Delta}{4m}.$$

Solving for $\tau$, we obtain that it suffices to set $\tau = \lceil 32\sigma^2 \log(4m/\delta) \cdot m^2/\Delta^2 \rceil$. Since we query each of the $m$ items $\tau$ times, the total number of queries is thus $m\lceil 32\sigma^2 \log(4m/\delta) \cdot m^2/\Delta^2 \rceil$.

# C. Proof of Theorem 2 (Upper Bound)

## C.1. Roadmap of the Proof

We will establish Theorem 2 by handling various cases separately, and along the way we will state more general results that hold for general choices of $\sigma$, allowing $\sigma = o(1)$ and $\sigma = \omega(1)$. The appendix is outlined as follows:

- In Section C.2, we study the regime in which there are at least as many queries as items (i.e., $q \geq m$), and to simplify the exposition we first focus on $\sigma = 1$. This part lays the main foundations for the subsequent parts.

- In Section C.3, we provide a straightforward extension from $\sigma = 1$ to general values of $\sigma$ when $q \geq m$.

- In Section C.4, we handle the case that $q < m$, this time turning immediately to general choices of $\sigma$.

Theorem 2 will then follow by combining Theorem 5 from Section C.3 (for $q \geq m$) with Corollary 1 from Section C.4 (for $q < m$). Recall that Section 6 discusses the case of general $\sigma$ in more detail, including comparing the upper and lower bounds.

## C.2. The Case $q \geq m$ and $\sigma = 1$

We first focus on the case that $\Delta$ grows sufficiently slowly with respect to $m$ such that $q \geq m$, i.e., there are enough queries to sample every item at least once. To reduce notation, we first focus on the case that $\sigma^2 = 1$; the value 1 could be replaced by any constant value, and this would only impact constant terms throughout the analysis. The case of general $\sigma$ (including $\sigma^2 = o(1)$ and $\sigma^2 = \omega(1)$ as $m \to \infty$) is deferred to Section C.3.

Formally, we will first prove the following.

**Theorem 4.** *In the case that $\sigma^2 = 1$, $\Delta \geq m^{1/4} \log^2 m$, and $\Delta = o\left(\frac{m}{\log m}\right)$,[8] there exists a polynomial-time algorithm that outputs an envy-free allocation with probability $1 - o(1)$ using the following number of queries:*

$$q = m\left\lceil 15\frac{m^{3/2}}{\Delta^2} \log m + \log^2 m \right\rceil. \tag{29}$$

*Moreover, these queries can be taken non-adaptively.*

### C.2.1. PROOF OF THEOREM 4 ($q \geq m$ AND $\sigma = 1$)

Recall that $u_i^a, u_i^b$ denote the true utilities, and $v_i^a, v_i^b$ denote the corresponding estimated utilities, namely

$$v_i^\nu = \frac{1}{\tau}\sum_{t=1}^{\tau} y_{i,t}^\nu, \quad \nu \in \{a, b\} \tag{30}$$

---

[8]This is a mild condition in view of the fact that $\Delta \leq m$, and more importantly, this result will only be used to establish Theorem 2 in the regime $\Delta = \widetilde{O}(m^{3/4})$. The regime of larger $\Delta$ will be handled via Theorem 6 below.

for $\tau = \frac{q}{m}$ independent observations $y_{i,t}^{\nu} \sim N(u_i^{\nu}, \sigma^2)$. Note that (29) ensures that $q$ is a multiple of $m$. Substituting $\sigma^2 = 1$ and using the fact that averaging $\tau$ i.i.d. Gaussians reduces the variance by a factor of $\tau$, we find that $v_i^{\nu} \sim N(u_i^{\nu}, m/q)$. We also note from (29) (and $\lceil x \rceil \geq x$) that

$$\frac{q}{m} \geq 15 \frac{m^{3/2}}{\Delta^2} \log m + \log^2 m. \tag{31}$$

Let $c > 0$ be a real constant (to be chosen later), and consider the following allocation rule:

$$\text{Assign item } i \text{ to Agent } \begin{cases} a & \text{if } cv_i^a - v_i^b > 0 \\ b & \text{otherwise.} \end{cases} \tag{32}$$

We see that the items more valuable to $a$ and less valuable to $b$ are assigned to $a$, and vice versa. This is done in a simple item-by-item manner, without any consideration for their "joint" behavior. By adjusting the constant $c$, we can adjust the "balance" of how many items are assigned to $a$ vs. $b$. As $c \to 0$, $b$ gets all items with positive $v_i^b$, and the effect of $v_i^a$ is diminished. Similarly, as $c \to \infty$, $a$ gets all items with positive $v_i^a$, and the effect of $v_i^b$ is diminished.

The analysis boils down to three main lemmas, which are stated below and summarized as follows:

- Lemma 7 states that a thresholding-based allocation captures "enough negative envy" in total.

- Lemma 9 states that we can make a "sufficiently balanced" allocation with respect to the observed (rather than true) valuations.

- Lemma 8 establishes concentration behavior sufficient to ensure that allocations that are (sufficiently) balanced with respect to the observed valuations are also balanced with respect to the true valuations.

We now proceed with formal statements. Let $e_a(c), e_b(c)$ be the envy for Agent $a$ and $b$ respectively under this allocation with respect to the true values $u_i^a, u_i^b$. Let $e_a'(c), e_b'(c)$ be defined similarly to $e_a(c)$ and $e_b(c)$, except that the estimates $v_i^a, v_i^b$ are used instead.

Mapping $\{a, b\}$ to $\{1, -1\}$, we define the following variable indicating the allocation of item $i$:

$$x_i(c) = \begin{cases} 1 & cv_i^a > v_i^b \\ -1 & cv_i^a \leq v_i^b, \end{cases} \tag{33}$$

which implies that

$$e_a(c) = -\sum_i x_i(c) u_i^a, \quad e_b(c) = \sum_i x_i(c) u_i^b, \tag{34}$$

$$e_a'(c) = -\sum_i x_i(c) v_i^a, \quad e_b'(c) = \sum_i x_i(c) v_i^b. \tag{35}$$

To avoid having to consider a continuum of $c$ values, we will confine $c$ to the following discrete set:

$$C = \left\{ \frac{k}{m^3} \;\middle|\; k = 1, 2, \ldots, m^6 \right\}. \tag{36}$$

We now formally state the three lemmas outlined above.

**Lemma 7.** *Under the setup of Theorem 4, let*

$$f(c) = \frac{ce_a(c) + e_b(c)}{1 + c}. \tag{37}$$

*Then with probability $1 - o(1)$, we have for all $c \in C$ that $f(c) \leq -\frac{1}{5} m^{-3/2} q^{1/2} \Delta^2 + \frac{m}{\sqrt{q}} \log m$.*

**Lemma 8.** *Under the setup of Theorem 4, let*

$$g(c) = \frac{1}{1 + c} (e_a(c) - e_a'(c) - ce_b(c) + ce_b'(c)). \tag{38}$$

*Then with probability $1 - o(1)$, we have for all $c \in C$ that $|g(c)| \leq \frac{m}{\sqrt{q}} \log m$.*

**Lemma 9.** *Under the setup of Theorem 4, let*

$$h(c) = \frac{1}{1+c}(e'_a(c) - ce'_b(c)). \tag{39}$$

*Then with probability $1 - o(1)$, there exists $c \in C$ such that*

$$\left(-\frac{2}{c} + 1\right)\frac{m}{\sqrt{q}}\log m \leq h(c) \leq (2c - 1)\frac{m}{\sqrt{q}}\log m. \tag{40}$$

Note that since $h(c)$ depends only on the observed valuations (not the true valuations), it is feasible for an algorithm to iterate over each $c \in C$ and check whether (40) holds. We let the algorithm use any such $c$ value.

**Proof of Theorem 4 given these lemmas.** Suppose that the conclusions of the three lemmas all hold, which is the case with probability $1 - o(1)$ by the union bound. Let $c$ be such that the conclusion of Lemma 9 holds. Combining (40) with Lemma 8, we obtain

$$-\frac{2}{c}\frac{m}{\sqrt{q}}\log m \leq \frac{e_a(c) - ce_b(c)}{1+c} = g(c) + h(c) \leq 2c\frac{m}{\sqrt{q}}\log m. \tag{41}$$

We then observe that

$$e_a(c) = \frac{1+c}{1+c^2}\left(cf(c) + \frac{e_a(c) - ce_b(c)}{1+c}\right) \tag{42}$$

$$\leq \frac{c(1+c)}{1+c^2}\left(-\frac{1}{5}m^{-3/2}q^{1/2}\Delta^2 + 3\frac{m}{\sqrt{q}}\log m\right) \tag{43}$$

$$\leq 0, \tag{44}$$

where:

- (42) follows by applying some simple manipulations via the definition of $f$ in (37);

- (43) follows by bounding $f(c)$ using Lemma 7 and bounding the second term using (41);

- (44) follows by multiplying the bracketed term by $\sqrt{q}$ and then applying $q \geq 15\frac{m^{5/2}}{\Delta^2}\log m$ (see (29)).

By analogous reasoning, we have

$$e_b(c) \overset{(37)}{=} \frac{1+c}{1+c^2}\left(f(c) - c \cdot \frac{e_a(c) - ce_b(c)}{1+c}\right) \overset{(41)}{\leq} \frac{1+c}{1+c^2}\left(-\frac{1}{5}m^{-3/2}q^{1/2}\Delta^2 + 3\frac{m}{\sqrt{q}}\log m\right) \overset{(29)}{\leq} 0. \tag{45}$$

Thus, we have constructed an envy-free allocation.

C.2.2. PROOF OF LEMMA 7 (BOUND ON $f$)

We start with some technical lemmas.

**Lemma 10.** *For all $c > 0$, we have $\sum_i |cu_i^a - u_i^b| \geq (1+c)\Delta$.*

*Proof.* Letting $\mathcal{A}^* = (\mathcal{A}_a^*, \mathcal{A}_b^*)$ be an optimal allocation, we have

$$\sum_i |cu_i^a - u_i^b| \geq \sum_{i \in \mathcal{A}_a^*}(cu_i^a - u_i^b) - \sum_{i \in \mathcal{A}_b^*}(cu_i^a - u_i^b) \tag{46}$$

$$= -c \cdot \text{Envy}_{a \to b}(\mathcal{A}^*) - \text{Envy}_{b \to a}(\mathcal{A}^*) \tag{47}$$

$$\geq (1+c)\Delta \tag{48}$$

as claimed. □

Next, we define the following quantity related to $f$:

$$z_i(c) = \sqrt{\frac{q}{m(1+c^2)}}(cu_i^a - u_i^b),\tag{49}$$

and let

$$Q(z) = \frac{1}{\sqrt{2\pi}}\int_z^\infty e^{-x^2/2}\, dx\tag{50}$$

be the upper tail of the standard Gaussian distribution. The following lemma motivates the definition of $z_i(c)$, and will be used later.

**Lemma 11.** *Under our allocation rule* (32) *and* $N(0,1)$ *noise, the probability that Agent $a$ gets item $i$ is $1 - Q(z_i(c))$.*

*Proof.* We established following (30) that $v_i^\nu \sim N(u_i^\nu, m/q)$ for $\nu \in \{a, b\}$, which implies that $cv_i^a - v_i^b$ follows a Gaussian distribution with mean $cu_i^a - u_i^b$ and variance $(1+c^2)\frac{m}{q}$, i.e., standard deviation $\sqrt{\frac{m(1+c^2)}{q}}$. To obtain the probability of this exceeding 0, we can simply subtract the mean and divide by the standard deviation to simplify to a tail bound for $N(0,1)$, and doing so gives a probability of $Q(-z_i(c)) = 1 - Q(z_i(c))$. $\qquad\square$

**Lemma 12.** *For all $c > 0$ and all sufficiently large $m$, we have*

$$\sum_i [1 - 2Q(z_i(c))]z_i(c) \geq 0.21\frac{q}{m^2}\frac{(1+c)^2}{1+c^2}\Delta^2.\tag{51}$$

*Proof.* In the proof of this lemma, we will simply write $z_i$ instead of $z_i(c)$, but the proof will hold for all $c$. If $z_i > 1$, then $1 - 2Q(z_i) \geq 1 - 2Q(1) > 0.68$, and therefore

$$(1 - 2Q(z_i))z_i \geq 0.68z_i.\tag{52}$$

On the other hand, if $0 \leq z_i \leq 1$, then

$$1 - 2Q(z_i) = \frac{1}{\sqrt{2\pi}}\int_{-z_i}^{z_i} e^{-x^2/2}\, dx \geq 2 \cdot \frac{1}{\sqrt{2\pi}} \cdot z_i \cdot e^{-1/2} \geq 0.48z_i,\tag{53}$$

and therefore

$$(1 - 2Q(z_i))z_i \geq 0.48z_i^2.\tag{54}$$

Since $(1 - 2Q(z_i))z_i$ is an even function, we conclude that

$$(1 - 2Q(z_i))z_i \geq \begin{cases} 0.68|z_i| & \text{if } |z_i| > 1 \\ 0.48z_i^2 & \text{if } |z_i| \leq 1. \end{cases}\tag{55}$$

Now, by Lemma 10, we have

$$\sum_i |z_i| = \sum_i \sqrt{\frac{q}{m(1+c^2)}}|cu_i^a - u_i^b| \geq \sqrt{\frac{q}{m}}\frac{1+c}{\sqrt{1+c^2}}\Delta.\tag{56}$$

We let

$$S_1 = \{i : |z_i| \leq 1\}, \quad S_2 = \{i : |z_i| > 1\}\tag{57}$$

and establish Lemma 12 via the following two cases, at least one of which must hold due to (56):

- *(Case 1: $\sum_{i \in S_1} |z_i| \geq \frac{2}{3}\sqrt{\frac{q}{m}}\frac{1+c}{\sqrt{1+c^2}}\Delta$.)* In this case, we have

$$\sum_i (1 - 2Q(z_i))z_i \geq \sum_{i \in S_1}(1 - 2Q(z_i))z_i \geq \sum_{i \in S_1} 0.48z_i^2 \geq 0.48\frac{(\sum_{i \in S_1}|z_i|)^2}{m} \geq 0.21\frac{q}{m^2}\frac{(1+c)^2}{1+c^2}\Delta^2,\tag{58}$$

where we used (54), the Cauchy–Schwarz inequality, and the assumption of this case.

- (*Case 2:* $\sum_{i \in S_2} |z_i| \geq \frac{1}{3} \sqrt{\frac{q}{m}} \frac{1+c}{\sqrt{1+c^2}} \Delta.$) In this case, we have

$$\sum_i (1 - 2Q(z_i))z_i \geq \sum_{i \in S_2} (1 - 2Q(z_i))z_i \geq 0.68 \sum_{i \in S_2} |z_i| \geq 0.21 \sqrt{\frac{q}{m}} \sqrt{\frac{(1+c)^2}{1+c^2}} \Delta. \tag{59}$$

It remains to show that

$$\sqrt{\frac{q}{m}} \sqrt{\frac{(1+c)^2}{1+c^2}} \Delta \geq \frac{q}{m^2} \frac{(1+c)^2}{1+c^2} \Delta^2, \tag{60}$$

which is equivalent to $q \leq \frac{m^3}{\Delta^2} \frac{1+c^2}{(1+c)^2}$. Noting that $\frac{1+c^2}{(1+c)^2} \in \left[\frac{1}{2}, 1\right]$ it suffices to show that

$$q \leq \frac{1}{2} \frac{m^3}{\Delta^2}, \tag{61}$$

which we establish via two cases:

- If the first term in (29) is dominant, then $q = \Theta(\frac{m^{5/2}}{\Delta^2} \log m)$, thus behaving as $o\left(\frac{m^3}{\Delta^2}\right)$.
- If the second term in (29) is dominant, then $q = \Theta(m \log^2 m)$, which again behaves as $o\left(\frac{m^3}{\Delta^2}\right)$ due to the assumption $\Delta = o\left(\frac{m}{\log m}\right)$ in Theorem 4.

Thus, in both cases we have $q \leq \frac{1}{2} \frac{m^3}{\Delta^2}$ when $m$ is large enough. $\qquad \square$

**Lemma 13.** *For any $c > 0$, we have with probability $1 - m^{-\omega(1)}$ that*[9]

$$f(c) \leq -\frac{1}{5} m^{-3/2} q^{1/2} \Delta^2 + \frac{m}{\sqrt{q}} \log m. \tag{62}$$

*Proof.* Recall the $\pm 1$-valued allocation indicator $x_i(c)$ from (33), the weighted difference of utilities $z_i(c)$ from (49), and $f(\cdot)$ from (37). These definitions imply that

$$f(c) = -\frac{\sum_i x_i(c)(cu_i^a - u_i^b)}{1 + c} = -\sqrt{\frac{m(1+c^2)}{q}} \sum_i \frac{x_i(c)z_i(c)}{1 + c}. \tag{63}$$

By the allocation probability established in Lemma 11, we have

$$\mathbb{E}[x_i(c)] = 1 - 2Q(z_i). \tag{64}$$

By linearity of expectation and Lemma 12, it follows from (63)–(64) that

$$\mathbb{E}[f(c)] = -\sqrt{\frac{m(1+c^2)}{q}} \sum_i \frac{1}{1+c}[1 - 2Q(z_i(c))]z_i(c) \leq -0.21 \frac{1+c}{\sqrt{1+c^2}} m^{-3/2} q^{1/2} \Delta^2 \leq -\frac{1}{5} m^{-3/2} q^{1/2} \Delta^2, \tag{65}$$

where the last step uses $\sqrt{1+c^2} \leq 1 + c$.

We now proceed to compute the corresponding variance. Since the $x_i(c)$ terms are independent of one another, we simply need to compute the variance of each term:

$$\mathrm{Var}[x_i(c)] = \mathbb{E}[x_i(c)^2] - \mathbb{E}[x_i(c)]^2 \overset{(64)}{=} 1 - [1 - 2Q(z_i(c))]^2 = 4Q(z_i(c))[1 - Q(z_i(c))] \tag{66}$$

so that

$$\mathrm{Var}[x_i(c)z_i(c)] = 4z_i(c)^2 Q(z_i(c))[1 - Q(z_i(c))]. \tag{67}$$

---

[9]Note that behaving as $m^{-\omega(1)}$ means decaying to zero faster than any polynomial in $m$.

Assuming momentarily that $z_i(c) \geq 0$, a standard upper bound on the Q-function (Mill's inequality) gives

$$Q(z_i(c)) \leq \frac{1}{\sqrt{2\pi}} \frac{e^{-\frac{1}{2}z_i(c)^2}}{z_i(c)}, \tag{68}$$

and hence

$$\text{Var}[x_i(c)z_i(c)] \overset{(67)}{=} 4z_i(c)^2 Q(z_i(c))[1 - Q(z_i(c))] \overset{(68)}{\leq} \frac{4}{\sqrt{2\pi}} e^{-\frac{1}{2}z_i(c)^2} z_i(c) \leq \frac{4}{\sqrt{2e\pi}} \leq 1, \tag{69}$$

where the second-last step uses $ze^{-z^2/2} \leq \frac{1}{\sqrt{e}}$, which can be verified by basic calculus (the maximum occurs at $z = 1$).

To drop the assumption $z_i(c) \geq 0$, we note that $\text{Var}[x_i(c)z_i(c)] = 4z_i(c)^2 Q(z_i(c))[1 - Q(z_i(c))]$ is an even function of $z_i(c)$ (due to $Q(-z) = 1 - Q(z)$), so that $\text{Var}(x_i(c)z_i(c)) \leq 1$ is still valid when $z_i(c) < 0$. Hence, (63) gives

$$\text{Var}[f(c)] = \frac{m(1 + c^2)}{q(1 + c)^2} \sum_i \text{Var}[x_i(c)z_i(c)] \leq \frac{m^2(1 + c^2)}{q(1 + c)^2} \leq \frac{m^2}{q}. \tag{70}$$

We are now in a position to apply Bernstein's inequality (see Lemma 2 in Appendix A), noting that the assumption $u_i \in [0, 1]$ implies $\left| x_i \cdot \frac{cu_i^a - u_i^b}{1+c} \right| \leq 1$, which in turn implies the centered counterpart $\left| (x_i - \mathbb{E}[x_i]) \cdot \frac{cu_i^a - u_i^b}{1+c} \right| \leq 2$ by the triangle inequality. Applying Lemma 2 (with parameters $T = m$, $b = 2$, and $\varepsilon = t/T$ therein) thus gives

$$\mathbb{P}[f(c) - \mathbb{E}[f(c)] > t] \leq 2\exp\left(-\frac{\frac{1}{2}t^2}{\text{Var}[f(c)] + \frac{2}{3}t}\right) \leq 2\exp\left(-\frac{\frac{1}{2}t^2}{\frac{m^2}{q} + \frac{2}{3}t}\right). \tag{71}$$

To simplify this, we set $t = \frac{m}{\sqrt{q}}\log m$ and write

$$\log^2 m \cdot \left(\frac{m^2}{q} + \frac{2}{3}t\right) \leq t^2 + \frac{2}{3}t^2 = \frac{5}{3}t^2, \tag{72}$$

where the inequality follows from the equivalence $t \log^2 m \leq t^2 \iff t \geq \log^2 m \iff \frac{m}{\sqrt{q}} \geq \log m \iff q \leq \frac{m^2}{\log^2 m}$, which in turn is seen to be true (for large enough $m$) by substituting the condition $\Delta \geq m^{1/4} \log^2 m$ from Theorem 4 into the choice of $q$ in (29). The inequality (72) then further implies

$$\frac{\frac{1}{2}t^2}{\frac{m^2}{q} + \frac{2}{3}t} \geq \frac{3}{10}\log^2 m. \tag{73}$$

Substituting into (71) and recalling the bound on $\mathbb{E}[f(c)]$ from (65), we obtain

$$\mathbb{P}\left[f(c) > -\frac{1}{5}m^{-3/2}q^{1/2}\Delta^2 + \frac{m}{\sqrt{q}}\log m\right] \leq 2\exp\left(-\frac{3}{10}\log^2 m\right) = m^{-\omega(1)}, \tag{74}$$

completing the proof. $\qquad \square$

Lemma 7 follows by taking a union bound over the $m^6$ values of $c \in C$ in Lemma 13.

### C.2.3. PROOF OF LEMMA 8 (BOUND ON $g$)

Recall that $g(\cdot)$ in (38) is expressed in terms of the quantities in (33)–(35). Substituting these definitions into (38), we can write $g(\cdot)$ as

$$g(c) = \frac{1}{1+c} \sum_i x_i(c)(v_i^a - u_i^a + c(v_i^b - u_i^b)). \tag{75}$$

To understand the behavior of $g$, the following covariance calculation will be useful:[10]

$$\text{Cov}(cv_i^a - v_i^b, v_i^a - u_i^a + c(v_i^b - u_i^b)) = \text{Cov}(cv_i^a, v_i^a - u_i^a) + \text{Cov}(-v_i^b, c(v_i^b - u_i^b)) \tag{76}$$

---
[10]Recall that $\text{Cov}[X, Y] = \mathbb{E}\big[(X - \mathbb{E}[X])(Y - \mathbb{E}[Y])\big]$

$$= c\mathrm{Var}[v_i^a] - c\mathrm{Var}[v_i^b] \tag{77}$$

$$= 0, \tag{78}$$

where (76) uses the fact that query outcomes for Agent $a$ and Agent $b$ are independent of one another, (77) uses the property that shifting by a constant (e.g., $u_i^a$) does not change the covariance, and (78) follows since the two estimated utilities are formed using the same number of samples with the same amount of noise. Since uncorrelated Gaussian random variables are also independent, we conclude that $(cv_i^a - v_i^b, v_i^a - u_i^a + c(v_i^b - u_i^b))$ form a pair of independent multivariate Gaussians.

Recall that we are considering a noise level of $\sigma^2 = 1$. By the independence of $(cv_i^a - v_i^b, v_i^a - u_i^a + c(v_i^b - u_i^b))$, conditioned on either $x_i(c) = 1$ or $x_i(c) = -1$ (each of which is deterministic given $cv_i^a - v_i^b$; see (32)), we have that $v_i^a - u_i^a + c(v_i^b - u_i^b)$ is a Gaussian distribution with zero mean and variance $\frac{m(1+c^2)}{q}$. It thus follows that $x_i(c)(v_i^a - u_i^a + c(v_i^b - u_i^b))$ is (unconditionally) a Gaussian distribution with zero mean and variance $\frac{m(1+c^2)}{q}$.

By the independence of query outcomes across items, we observe that $g(c)$ is a sum of $m$ independent Gaussians, and is thus itself Gaussian, with zero mean and variance $\frac{m^2}{q}\frac{1+c^2}{(1+c)^2}$, i.e., standard deviation $\frac{m}{\sqrt{q}}\frac{\sqrt{1+c^2}}{1+c}$. Since $\frac{\sqrt{1+c^2}}{1+c} = \Theta(1)$ regardless of $c$, it follows that

$$\mathbb{P}\left[|g(c)| \geq \frac{m}{\sqrt{q}}\log m\right] \leq \exp\left(-\Omega((\log m)^2)\right) = m^{-\omega(1)}. \tag{79}$$

Taking a union bound over the $m^6$ values of $c \in C$ gives the desired result.

### C.2.4. PROOF OF LEMMA 9 (BOUND ON $h$)

We start with the following useful lemma regarding the discrete set $C$ (see (36)).

**Lemma 14.** *With probability* $1 - o(1)$, *for all pairs* $(c_1, c_2)$ *of consecutive elements in $C$, there exist at most* $\log m$ *items $i$ with* $x_i(c_1) \neq x_i(c_2)$.

*Proof.* By the allocation rule in (32), for each $i$, the condition $x_i(c_1) \neq x_i(c_2)$ implies that $v_i^b$ is between $c_1 v_i^a$ and $c_2 v_i^a$; note that $c_2 - c_1 = \frac{1}{m^3}$ (see (36)). It is also useful to note the following consequence of $u_i^a \in [0, 1]$ and $v_i^a \sim N(u_i^a, m/q)$:

$$\mathbb{P}[v_i^a \geq 2] = \mathbb{P}\left[N(0,1) > \frac{2 - u_i^a}{\sqrt{m/q}}\right] \leq \mathbb{P}\left[N(0,1) > \sqrt{q/m}\right] \leq \mathbb{P}[N(0,1) > \log m], \tag{80}$$

where the last step uses $q/m \geq \log^2 m$ (see (29)). This decays as $\exp\left(-\Omega((\log m)^2)\right) = m^{-\omega(1)}$, i.e., faster than polynomial in $m$. An entirely analogous argument holds for $\mathbb{P}[v_i^a \leq -2]$, and combining these gives

$$\mathbb{P}[|v_i^a| \geq 2] = m^{-\omega(1)}. \tag{81}$$

Next, we observe that

$$\mathbb{P}[v_i^b \in [c_1 v_i^a, c_2 v_i^a]] = \frac{1}{\sqrt{2\pi m/q}}\left|\int_{c_1 v_i^a}^{c_2 v_i^a} e^{-\frac{q}{m}(x - u_i^b)^2/2}\,dx\right| \leq \sqrt{\frac{q}{m}}|v_i^a|(c_2 - c_1). \tag{82}$$

The three terms on the right-hand side are respectively upper bounded by $m$ (by crudely loosening (29) to $q \leq m^3$), 2 (under the high-probability event in (81)), and $\frac{1}{m^3}$, yielding an overall upper bound of $2m^{-2}$. The probability of having at least $\log m$ such items is thus upper bounded by

$$\binom{m}{\log m}(2m^{-2})^{\log m} \leq m^{\log m}(2m^{-2})^{\log m} = m^{-\omega(1)}. \tag{83}$$

Taking the union bound over all $m^6 - 1$ possible pairs $(c_1, c_2)$ gives the desired result. $\qquad\square$

We now break down the desired inequality (40) into the upper bound and lower bound separately:

$$h(c) \leq (2c - 1)\frac{m}{\sqrt{q}}\log m, \tag{84}$$

$$h(c) \geq \left( -\frac{2}{c} + 1 \right) \frac{m}{\sqrt{q}} \log m. \tag{85}$$

Recall $e'(c)$ from (35), and observe that for all $c$, under the high-probability event $|v_i| \leq 2$, we have

$$|h(c)| \leq \frac{1}{1+c}(|e'_a(c)| + c|e'_b(c)|) \leq |e'_a(c)| + |e'_b(c)| \leq \sum_i |v_i^a| + \sum_i |v_i^b| \leq 4m. \tag{86}$$

When $c$ takes its highest value within $C$ (i.e., $c = m^3$), we have $h(c) \leq 4m \leq (2c - 1)\frac{m}{\sqrt{q}} \log m$ (e.g., even crudely using $q \leq m^3$) so that (84) holds. On the other hand, when $c$ takes its smallest value (i.e., $c = \frac{1}{m^3}$), we have $h(c) \geq -4m \geq \left( -\frac{2}{c} + 1 \right) \frac{m}{\sqrt{q}} \log m$ so that (85) holds.

Suppose for contradiction that (84) and (85) never hold true at the same time. Let $c_2$ be the smallest $c \in C$ such that (84) holds (since $c = m^3$ satisfies (84), such a $c$ exists and the smallest is well defined). Note that $c_2 \neq \frac{1}{m^3}$, since (85) holds when $c = \frac{1}{m^3}$. Let $c_1 = c_2 - \frac{1}{m^3} \in C$. By the definition of $c_2$ and what we assumed (for contradiction), (85) fails for $c_2$ and (84) fails for $c_1 = c_2 - \frac{1}{m^3}$. We will bound $h(c_2) - h(c_1)$ in two ways and show that this is impossible.

Since (85) fails for $c_2$ and (84) fails for $c_1$, we have

$$h(c_2) - h(c_1) \leq \left( -\frac{2}{c_2} + 1 \right) \frac{m}{\sqrt{q}} \log m - (2c_1 - 1)\frac{m}{\sqrt{q}} \log m \tag{87}$$

$$= \left( -\frac{2}{c_2} + 1 \right) \frac{m}{\sqrt{q}} \log m - \left( 2c_2 - \frac{2}{m^3} - 1 \right) \frac{m}{\sqrt{q}} \log m \tag{88}$$

$$\leq -\frac{m}{\sqrt{q}} \log m, \tag{89}$$

where we used $x + \frac{1}{x} \geq 2$ and $\frac{2}{m^3} \leq 1$.

On the other hand, by the definition of $h$ in (39), we have

$$h(c_2) - h(c_1) = \frac{1}{1+c_2}(e'_a(c_2) - c_2 e'_b(c_2)) - \frac{1}{1+c_1}(e'_a(c_1) - c_1 e'_b(c_1)) \tag{90}$$

$$= \left( \frac{1}{1+c_2}(e'_a(c_2) - c_2 e'_b(c_2)) - \frac{1}{1+c_1}(e'_a(c_2) - c_1 e'_b(c_2)) \right)$$

$$+ \left( \frac{1}{1+c_1}(e'_a(c_2) - c_1 e'_b(c_2)) - \frac{1}{1+c_1}(e'_a(c_1) - c_1 e'_b(c_1)) \right) \tag{91}$$

$$= \frac{(c_1 - c_2)(e'_a(c_2) + e'_b(c_2))}{(1+c_1)(1+c_2)} + \frac{1}{1+c_1}[e'_a(c_2) - e'_a(c_1) - c_1(e'_b(c_2) - e'_b(c_1))]. \tag{92}$$

We proceed to handle the two terms on the right-hand side:

- For the second term, under the high-probability events (i) at most $\log m$ items have $x_i(c_1) \neq x_i(c_2)$ (Lemma 21), and (ii) $|v_i^a| \leq 2$ and $|v_i^b| \leq 2$ (see (81)), we have

$$|e'_a(c_2) - e'_a(c_1) - c_1(e'_b(c_2) - e'_b(c_1))| \leq 2(1 + c_1) \log m, \tag{93}$$

  and this $1 + c_1$ term cancels with $\frac{1}{1+c_1}$ in (92).

- For the first term, we again use $|v_i^a| \leq 2$ and $|v_i^b| \leq 2$, but more crudely bound

$$|e'_a(c_2) + e'_b(c_2)| \leq 4m \tag{94}$$

  since each $e'_\nu$ ($\nu \in \{a, b\}$) consists of a summation over $m$ estimated utilities (weighted by $+1$ or $-1$). It follows that

$$\left| \frac{(c_1 - c_2)(e'_a(c_2) + e'_b(c_2))}{(1+c_1)(1+c_2)} \right| \leq \frac{1}{m^3} \cdot 4m = \frac{4}{m^2}. \tag{95}$$

Combining these findings into (92), we have with probability $1 - o(1)$ that

$$|h(c_2) - h(c_1)| \le \frac{4}{m^2} + 2\log m \tag{96}$$

which we claim implies for sufficiently large $m$ that

$$|h(c_2) - h(c_1)| < \frac{m}{\sqrt{q}} \log m. \tag{97}$$

To see that (96) implies (97), we substitute the assumption $\Delta \ge m^{1/4} \log^2 m$ from Theorem 4 into the choice of $q$ in (29) to obtain $q \le m \lceil \frac{15m}{\log^3 m} + \log^2 m \rceil$, which scales as $o(m^2)$, thus implying $\frac{m}{\sqrt{q}} \log m = \omega(\log m)$.

Equations (89) and (97) directly contradict one another, so our assumption that no such $c \in C$ exists must be false. Therefore, there must exist some $c \in C$ such that both (84) and (85) hold.

### C.3. The Case $q \ge m$ with General $\sigma$

The following theorem generalizes Theorem 4 from $\sigma = 1$ to general choices of $\sigma$.

**Theorem 5.** *If $\Delta \ge m^{1/4} \log^2 m$ and $\Delta = o\left(\frac{m}{\log m}\right)$, then there exists a polynomial-time algorithm that outputs an envy-free allocation with probability $1 - o(1)$ using the following number of queries:*

$$q = m \left\lceil \sigma^2 \left( 15 \frac{m^{3/2}}{\Delta^2} \log m + \log^2 m \right) \right\rceil. \tag{98}$$

*Proof.* The proof is nearly identical to that of Theorem 4. Recall that in the proof of Theorem 4, each item is sampled $q/m$ times, so that $v_i \sim N(u_i, m/q)$ (omitting the superscript $a$ or $b$ for brevity). Here we again sample each item $\frac{q}{m}$ times, and consider the following two cases:

- If $q > m$, then $v_i \sim N(u_i, \sigma^2 m/q)$. When the argument to $\lceil \cdot \rceil$ in (98) is already an integer, this evaluates to $N(u_i, \frac{\Delta^2}{15m^{3/2}\log m + \Delta^2 \log^2 m})$, in which the variance of $v_i$ does not depend on $\sigma$ and thus the proof for $\sigma = 1$ applies. Slightly more care is needed when there is rounding involved, and we avoid repeating the details, but instead simply provide the intuition that rounding up to a multiple of $m$ is inconsequential when $q > m$, since it only amounts to multiplying by a factor of 2 or less.

- If $q = m$, then each item is sampled exactly once. In this case, we observe that the number of samples equaling $m$ in (98) implies that $\sigma^2 \le \frac{\Delta^2}{15m^{3/2}\log m + \Delta^2 \log^2 m}$, which means that extra noise may be added to produce $v_i \sim N(u_i, \frac{\Delta^2}{15m^{3/2}\log m + \Delta^2 \log^2 m})$ (instead of $N(u_i, \sigma^2)$) and thus recover the same conditions as the case $q > m$. $\square$

Observe that setting $\sigma = 1$ in Theorem 5 recovers Theorem 4. Moreover, whenever $\Delta \le \widetilde{O}(m^{3/4})$, the bound on $q$ in (98) behaves as $\widetilde{O}\left(\frac{m^{2.5}}{\Delta^2}\right)$, thus recovering Theorem 2 in this regime. The regime $\Delta \gg m^{3/4}$ will be handled in the next subsection.

### C.4. The Case $q < m$

In the case that $q < m$ (i.e., fewer queries than items), handling general $\sigma$ turns out to require non-trivial additional effort compared to $\sigma = 1$, so to avoid repetition we handle general $\sigma$ from the beginning. Using Theorem 4 as a stepping stone, we will show the following.

**Theorem 6.** *Suppose that the following conditions hold:*

$$\Delta^2 > 160\sigma m^{3/2} \log^2 m \tag{99}$$

$$\Delta^4 > 160^2 m^3 \sigma^4 \log^2 m \tag{100}$$

$$\Delta < \min\{2\sigma^2 m, \sqrt{2}\sigma m\}. \tag{101}$$

*Then, there exists a polynomial-time algorithm that outputs an envy-free allocation with probability $1 - o(1)$ using the following number of queries:*

$$q = \left\lceil \max \left\{ 160^2 \frac{m^4}{\Delta^4} \sigma^4 \log^2 m, \ 160 \frac{\sigma m^{5/2}}{\Delta^2} \log^2 m \right\} \right\rceil. \tag{102}$$

Note that by substituting (99) and (100) into (102), we indeed find that $q \leq m$ (with strict inequality except in some very specific cases). We also note that squaring both sides of $\Delta < 2\sigma^2 m$ (from (101)), solving for $\sigma^4$, and substituting into (100), we obtain

$$\Delta > 80\sqrt{m} \log m \tag{103}$$

which provides an explicit lower bound on $\Delta$ not depending on $\sigma$. We remark that we have made no attempt to optimize the constant factors in our results, as our focus is on the scaling laws.

Next, we state the following simplified corollary for the case of a constant noise level.

**Corollary 1.** *In the case that $\sigma^2 > 0$ is a constant (not depending on $m$) and we have $\Delta > Cm^{3/4} \log m$ and $\Delta < C'm$ for sufficiently large $C$ and sufficiently small $C'$, there exists a polynomial-time algorithm that outputs an envy-free allocation with probability $1 - o(1)$ using the following number of non-adaptive queries:*

$$q = \left\lceil 160 \frac{\sigma m^{5/2}}{\Delta^2} \log^2 m \right\rceil. \tag{104}$$

*Proof.* The assumption $\Delta > Cm^{3/4} \log m$ for sufficiently large $C$, along with $\sigma$ being a constant, implies that conditions (99) and (100) are satisfied. Condition (101) also holds by setting $C' < \min\{2\sigma^2, \sqrt{2}\sigma\}$. Finally, the ratio between the two terms in (102) is $160\sigma^3 \frac{m^{3/2}}{\Delta^2}$, so the assumption $\Delta > Cm^{3/4} \log m$ implies (for suitable $C$ and large enough $m$) that the second term achieves the maximum in (102), thus recovering (104). $\qquad\square$

This corollary recovers Theorem 2 (stated less precisely using $\widetilde{O}(\cdot)$ notation) in the regime $\Delta \geq \omega(m^{3/4} \log m)$. Since we recovered Theorem 2 for any $\Delta \leq \widetilde{O}(m^{3/4})$ in the previous subsection, this means that we have now recovered Theorem 2 in its entirety.

### C.4.1. PROOF OF THEOREM 6 VIA AUXILIARY RESULTS

We consider randomly querying $q < m$ items, uniformly randomly across all $\binom{m}{q}$ subsets, once each. We do not query the remaining $m - q$ items. Let $S$ denote the set of queried items, meaning its complement $S^c = \{1, \ldots, m\} \setminus S$ is the set of non-queried items.

We again let

$$C = \left\{ \frac{k}{m^3} \ \middle| \ k = 1, 2, \ldots m^6 \right\}. \tag{105}$$

For some $c \in C$ to be chosen later, for the items that we sample, we follow our earlier strategy:

$$\text{Assign item } i \text{ to agent} \begin{cases} a & \text{if } cv_i^a - v_i^b > 0 \\ b & \text{otherwise.} \end{cases} \tag{106}$$

On the other hand, for the items we do not sample, we assign them to each agent randomly with probability $\frac{1}{2}$ each, independent of all other choices.

It will be useful to decompose the envy quantities from (34)–(35) into the contributions of the queried items $S$ and non-queried items $S^c$:

$$e_a(c) = \underbrace{-\sum_{i \in S} x_i(c)u_i^a}_{\bar{e}_a(c)} \underbrace{-\sum_{i \in S^c} x_i(c)u_i^a}_{\tilde{e}_a(c)}, \quad e_b(c) = \underbrace{\sum_{i \in S} x_i(c)u_i^b}_{\bar{e}_b(c)} + \underbrace{\sum_{i \in S^c} x_i(c)u_i^b}_{\tilde{e}_b(c)}, \tag{107}$$

$$e'_a(c) = \underbrace{-\sum_{i \in S} x_i(c)v_i^a}_{\overline{e}'_a(c)} \underbrace{- \sum_{i \in S^c} x_i(c)v_i^a}_{\widetilde{e}'_a(c)}, \quad e'_b(c) = \underbrace{\sum_{i \in S} x_i(c)v_i^b}_{\overline{e}'_b(c)} + \underbrace{\sum_{i \in S^c} x_i(c)v_i^b}_{\widetilde{e}'_b(c)}. \tag{108}$$

In short, we use $\overline{e}$ for the queried part, $\widetilde{e}$ for the non-queried part, and $e$ for the total.

Let $\overline{f}(c), \overline{g}(c), \overline{h}(c)$ be defined similarly to $f(c), g(c), h(c)$ in (37)–(39), except that we include only the items that we query:

$$\overline{f}(c) = \frac{c\overline{e}_a(c) + \overline{e}_b(c)}{1 + c}, \tag{109}$$

$$\overline{g}(c) = \frac{1}{1 + c}(\overline{e}_a(c) - \overline{e}'_a(c) - c\overline{e}_b(c) + c\overline{e}'_b(c)) \tag{110}$$

$$\overline{h}(c) = \frac{1}{1 + c}(\overline{e}'_a(c) - c\overline{e}'_b(c)). \tag{111}$$

We then have the following analogs of our earlier lemmas.

**Lemma 15.** (Analog of Lemma 7) *Under the setup of Theorem 6, with probability $1 - o(1)$, we have*

$$\overline{f}(c) \leq -\frac{1}{20}\frac{q\Delta^2}{\sigma m^2} + \sigma\sqrt{q}\log q \tag{112}$$

*for all $c \in C$.*

**Lemma 16.** (Analog of Lemma 8) *Under the setup of Theorem 6, with probability $1 - o(1)$, we have for all $c \in C$ that $|\overline{g}(c)| \leq \sigma\sqrt{q}\log q$.*

*Proof.* We first consider a fixed choice of $c \in C$. Observe that for each queried item, $v_i^a - u_i^a + c(v_i^b - u_i^b)$ is a Gaussian distribution with zero mean and variance $\sigma^2(1 + c^2)$. Hence, $\overline{g}(c)$ is a Gaussian distribution with zero mean and variance $\sigma^2 q \frac{1+c^2}{(1+c)^2}$, i.e., standard deviation $\sigma\sqrt{q}\frac{\sqrt{1+c^2}}{1+c}$. Since $\frac{\sqrt{1+c^2}}{1+c} \leq 1$, this means that $\mathbb{P}[|\overline{g}(c)| \geq \sigma\sqrt{q}\log q]$ is a Gaussian tail bound of at least $\log q$ standard deviations from the mean, and is thus superpolynomially small. Applying a union bound over the $m^6$ values of $c \in C$ completes the proof. $\square$

**Lemma 17.** (Analog of Lemma 9) *Under the setup of Theorem 6, with probability $1 - o(1)$, there exists $c \in C$ such that*

$$\left(-\frac{3}{c} + 2\right)(\sqrt{m}\log^2 m + \sigma\sqrt{q}\log m) \leq \overline{h}(c) \leq (3c - 2)(\sqrt{m}\log^2 m + \sigma\sqrt{q}\log m). \tag{113}$$

Once again, since $\overline{h}(c)$ depends only on the estimated utilities, it is feasible for an algorithm to iterate over each $c \in C$ and check whether (113) holds. We let the algorithm use any such $c$ value.

**Proof of Theorem 6 given these lemmas.** Along with $\overline{f}(c)$ from (109), we consider the following counterparts with non-queried items only, and with all items:

$$\widetilde{f}(c) = \frac{c\widetilde{e}_a(c) + \widetilde{e}_b(c)}{1 + c}, \quad f(c) = \frac{ce_a(c) + e_b(c)}{1 + c}, \tag{114}$$

thus yielding $f(c) = \overline{f}(c) + \widetilde{f}(c)$. We also define $\widetilde{g}(c), g(c)$ and $\widetilde{h}(c), h(c)$ in an analogous manner. The following lemma bounds various contributions from the non-queried items.

**Lemma 18.** *With probability $1 - o(1)$, it holds for all $c \in C$ that*

$$|\widetilde{f}(c)| \leq \sqrt{m}\log m, \tag{115}$$

$$\left|\frac{\widetilde{e}_a(c) - c\widetilde{e}_b(c)}{1 + c}\right| \leq \sqrt{m}\log m. \tag{116}$$

*Proof.* We first consider a fixed choice of $c \in C$. Recall that the non-queried items are assigned independently with probability $\frac{1}{2}$ each. Hence, from (107), $\widetilde{e}_\nu(c)$ (for $\nu \in a, b$) is a summation of $|S^c| \leq m$ independent random variables, each taking two values of the form $\{u, -u\}$ (for some $u \in [0, 1]$) with probability $\frac{1}{2}$ each. We can thus apply Hoeffding's inequality to obtain

$$\mathbb{P}[|\widetilde{e}_\nu(c)| \geq t] \leq 2 \exp\left(\frac{-2t^2}{4m}\right). \tag{117}$$

Setting $t = \sqrt{m} \log m$, we find that this decays as $m^{-\omega(1)}$. Hence, by a union bound over $c \in C$ and $\nu \in \{a, b\}$, we have $|\widetilde{e}_\nu(c)| \leq \sqrt{m} \log m$ for all $(c, \nu)$ with probability $1 - o(1)$. The first claim of the lemma follows by substituting into the definition of $\widetilde{f}$ in (114), and the second claim follows similarly. $\square$

In the following, we condition on the high-probability events of Lemmas 15, 16, 17, and 18, for a combined probability of $1 - o(1)$.

Let $c$ be such that the conclusion of Lemma 17 holds. Towards understanding the envy of the allocation, we first write

$$\frac{e_a(c) - ce_b(c)}{1+c} = \underbrace{\frac{\overline{e}_a(c) - c\overline{e}_b(c)}{1+c}}_{=\overline{g}(c)+\overline{h}(c)} + \underbrace{\frac{\widetilde{e}_a(c) - c\widetilde{e}_b(c)}{1+c}}_{\text{Bounded in Lemma 18}} \tag{118}$$

$$\leq \overline{g}(c) + \overline{h}(c) + \sqrt{m} \log m. \tag{119}$$

Further applying $|\overline{g}(c)| \leq \sigma\sqrt{q} \log m$ (via Lemma 16 and $q \leq m$) and $\overline{h}(c) \leq (3c - 2)(\sqrt{m} \log^2 m + \sigma\sqrt{q} \log m)$ (via Lemma 17), we obtain

$$\frac{e_a(c) - ce_b(c)}{1+c} \leq 3c(\sqrt{m} \log^2 m + \sigma\sqrt{q} \log m). \tag{120}$$

A similar argument gives the lower bound

$$\frac{e_a(c) - ce_b(c)}{1+c} \geq -\frac{3}{c}(\sqrt{m} \log^2 m + \sigma\sqrt{q} \log m). \tag{121}$$

We now observe that

$$e_a(c) = \frac{1+c}{1+c^2}\left(cf(c) + \frac{e_a(c) - ce_b(c)}{1+c}\right) \tag{122}$$

$$\leq \frac{c(1+c)}{1+c^2}\left(-\frac{1}{20}\frac{q\Delta^2}{\sigma m^2} + \sqrt{m} \log m + \sigma\sqrt{q} \log m + 3\sqrt{m} \log^2 m + 3\sigma\sqrt{q} \log m\right) \tag{123}$$

$$\leq 0, \tag{124}$$

where:

- (122) follows by re-arranging the definition of $f$ from (114);

- (123) follows by applying $f = \overline{f} + \widetilde{f}$ along with Lemmas 15 and 18 and $q \leq m$ for the first term, and applying (120) to the second term;

- (124) follows by crudely bounding $\sqrt{m} \log m \leq \sqrt{m} \log^2 m$ and then adding the following two inequalities:

$$-\frac{1}{40}\frac{q\Delta^2}{\sigma m^2} + 4\sigma\sqrt{q} \log m \leq 0 \tag{125}$$

$$-\frac{1}{40}\frac{q\Delta^2}{\sigma m^2} + 4\sqrt{m} \log^2 m \leq 0, \tag{126}$$

which in turn follow from the fact that $q$ is lower bounded by each of the two terms in (102). (In (125) we can first simplify by dividing through by $\sqrt{q}$.)

By analogous reasoning (but with (121) instead of (120)), we have

$$e_b(c) = \frac{1+c}{1+c^2}\left(f(c) - c \cdot \frac{e_a(c) - ce_b(c)}{1+c}\right)$$

$$\leq \frac{1+c}{1+c^2}\left(-\frac{1}{20}\frac{q\Delta^2}{\sigma m^2} + \sqrt{m}\log m + \sigma\sqrt{q}\log m + 3\sqrt{m}\log^2 m + 3\sigma\sqrt{q}\log m\right) \leq 0. \tag{127}$$

Thus, we have constructed an envy-free allocation.

### C.4.2. PROOF OF LEMMA 15 (BOUND ON $\overline{f}$)

We will focus on a single value of $c$ and then generalize. Throughout the analysis, we let $\Delta' = \frac{1}{2}\frac{q}{m}\Delta$, which measures the gap of the "smaller problem" consisting of only $q < m$ queried items. The factor of $\frac{1}{2}$ is to ensure that the gap is valid with high probability, as formalized in the following lemma.

**Lemma 19.** (Analog of Lemma 10) *Under the setup of Theorem 6, for any $c > 0$, we have probability $1 - m^{-\omega(1)}$ that*

$$\sum_{i \in S} |cu_i^a - u_i^b| \geq (1+c)\Delta'. \tag{128}$$

*Proof.* By Lemma 10, we have $\sum_i |cu_i^a - u_i^b| \geq (1+c)\Delta$. We compute the following mean, noting that each item is in $S$ with probability $\frac{q}{m}$:

$$\mathbb{E}\left[\sum_{i \in S} |cu_i^a - u_i^b|\right] = \frac{q}{m}\sum_i |cu_i^a - u_i^b| \geq \frac{q}{m}(1+c)\Delta = 2(1+c)\Delta'. \tag{129}$$

We now observe that $\sum_{i \in S} |cu_i^a - u_i^b|$ is equal to the sum of values upon drawing $q$ items from a population of size $m$ without replacement, where the $i$-th value is $|cu_i^a - u_i^b|$. This allows us to apply Hoeffding's inequality for sampling without replacement (e.g., see (Bardenet & Maillard, 2015, Prop. 1.2)) to obtain

$$\mathbb{P}\left[\sum_{i \in S} |cu_i^a - u_i^b| \leq (1+c)\Delta'\right] \leq \mathbb{P}\left[\sum_{i \in S} |cu_i^a - u_i^b| \leq \mathbb{E}\left[\sum_{i \in S} |cu_i^a - u_i^b|\right] - (1+c)\Delta'\right] \tag{130}$$

$$\leq \exp\left(-\frac{2\big((1+c)\Delta'\big)^2}{q(1+c)^2}\right) \tag{131}$$

$$= \exp(-2(\Delta')^2/q) \tag{132}$$

$$\leq \exp(-2(\Delta')^2/m) \tag{133}$$

$$= \exp\left(-\frac{1}{2}\frac{q^2\Delta^2}{m^3}\right), \tag{134}$$

where in (133) we applied $q \leq m$, and in (134) we substituted $\Delta' = \frac{1}{2}\frac{q}{m}\Delta$. Using $q \geq 160\frac{\sigma m^{5/2}}{\Delta^2}\log^2 m$ from (102) followed by $\sigma > \frac{\Delta}{\sqrt{2}m}$ from (101) and squaring, we have $q^2 \geq \frac{160^2 m^3 \log^4 m}{2\Delta^2}$, which implies that $\frac{1}{2}\frac{q^2\Delta^2}{m^3} = \Omega(\log^4 m)$ and thus (134) decays to zero superpolynomially fast in $m$. $\square$

In the proof of Theorem 4, the only way we used the existence of an allocation with envy at most $-\Delta$ was through Lemma 10. As a result, Lemma 19 can be used to prove the following analog of Lemma 13, with $q$ in place of $m$ (since we are working with the $q < m$ queried items) and $\Delta'$ in place of $\Delta$ (since Lemma 19 justifies $\Delta'$ as the "effective gap" for the reduced problem).

**Lemma 20.** (Analog of Lemma 13) *Under the setup of Theorem 6, for any $c > 0$, it holds with probability $1 - m^{-\omega(1)}$ that*

$$\overline{f}(c) \leq -\frac{1}{20}\frac{q\Delta^2}{\sigma m^2} + \sigma\sqrt{q}\log q. \tag{135}$$

*Proof.* The proof is mostly the same as the proof of Lemma 13, so we only describe the differences. As hinted above, we substitute different choices of variables:

- We set $m' = q$ because here we are only working with the $q$ queried items;

- We set $\Delta' = \frac{1}{2}\frac{q}{m}\Delta$ in accordance with Lemma 19;

- We set $q' = \frac{q}{\sigma^2}$ to align with the fact that averaging $\frac{\tau}{\sigma^2}$ unit-variance Gaussians yields the same variance as averaging $\tau$ variance-$\sigma^2$ Gaussians. While this explanation may suggest that $q'$ should be an integer, this is not the case – we are still sampling every item once with $N(0, \sigma^2)$ noise, and it is the *difference in noise level* that leads to each occurrence of $q$ ultimately being replaced by $\frac{q}{\sigma^2}$. Specifically, it is easy to check that for Lemma 11 to remain true as stated, we should replace $z_i(c) = \frac{1}{\sqrt{\sigma^2(1+c^2)}}(cu_i^a - u_i^b)$ in (49). This amounts to replacing $\frac{m'}{q'}$ by $\sigma^2$, from which we get $q' = \frac{q}{\sigma^2}$ upon substituting $m' = q$.

Lemma 13 is proved via Lemma 12, whose proof is essentially unchanged except that we need to check that (61) holds upon the above-given variable substitutions (i.e., $q' \le \frac{1}{2}\frac{(m')^3}{(\Delta')^2}$). We have

$$q' \le \frac{1}{2}\frac{(m')^3}{(\Delta')^2} \iff \frac{q}{\sigma^2} \le \frac{q^3}{2(\frac{1}{2}\frac{q}{m}\Delta)^2} \iff \Delta \le \sqrt{2}\sigma m, \tag{136}$$

which is satisfied due to (101). The proof of Lemma 13 can then be repeated, with the above variable substitutions yielding the following analog of (62):

$$\overline{f}(c) \le -\frac{1}{5}(m')^{-3/2}(q')^{1/2}(\Delta')^2 + \frac{m'}{\sqrt{q'}}\log m' = -\frac{1}{20}\frac{q\Delta^2}{\sigma m^2} + \sigma\sqrt{q}\log q. \tag{137}$$

This completes the proof. $\square$

A union bound over the $m^6$ values of $c \in C$ then gives Lemma 15.

### C.4.3. PROOF OF LEMMA 17 (BOUND ON $\overline{h}$)

Recall that $x_i(c)$ denotes the $\pm 1$-valued allocation indicator variable as defined in (33), and that the set $C$ is defined in (105).

**Lemma 21.** (Analog of Lemma 14) *Under the setup of Theorem 6, with probability $1 - o(1)$, for all pairs $(c_1, c_2)$ of consecutive elements in $C$, there exist at most $\log m$ items $i$ with $x_i(c_1) \ne x_i(c_2)$.*

*Proof.* We first establish some loose but useful bounds on $\sigma$. For an upper bound, using (100) along with $\Delta \le m$ and $160^2\log^2 m \ge 1$ gives

$$\sigma \le m^{1/4}. \tag{138}$$

For a lower bound, using the first term in (101) followed by the crude bound $\Delta \ge 2$ (which follows from the much stronger bound in (103)) gives

$$\sigma \ge \sqrt{\frac{\Delta}{2m}} \ge \frac{1}{\sqrt{m}}. \tag{139}$$

For each $i$, the condition $x_i(c_1) \ne x_i(c_2)$ implies that $v_i^b$ lies between $c_1v_i^a$ and $c_2v_i^a$. Observe that combining $u_i^a \in [0, 1]$ and $v_i \sim N(u_i^a, \sigma^2)$ gives

$$\mathbb{P}\left[|v_i^a| \ge \sqrt{m}\log m\right] = 2\mathbb{P}\left[N(0, 1) > \frac{\sqrt{m}\log m - u_i^a}{\sigma}\right] \overset{(138)}{\le} \mathbb{P}\left[N(0, 1) > \Omega(m^{1/4}\log m)\right]. \tag{140}$$

This decays as $\exp\left(-\Omega\left(\sqrt{m}(\log m)^2\right)\right)$, which is strictly faster than any polynomial in $m$.

Under the event $|v_i^a| \le \sqrt{m}\log m$, we have for any consecutive $(c_1, c_2)$ (i.e., $c_2 = c_1 + \frac{1}{m^3}$) that

$$\mathbb{P}\left[v_i^b \text{ between } c_1v_i^a \text{ and } c_2v_i^a\right] = \frac{1}{\sqrt{2\pi\sigma^2}}\left|\int_{c_1v_i^a}^{c_2v_i^a} e^{-(x-u_i^b)^2/(2\sigma^2)}\,dx\right| \le \frac{1}{\sigma}|v_i^a|(c_2 - c_1) \le m^{-2}\log m, \tag{141}$$

where in the last step, the three terms are bounded by $\sqrt{m}$ (see (139)), $\sqrt{m}\log m$, and $\frac{1}{m^3}$ respectively. By independence across items, the probability of having at least $\log m$ items with the above property is upper bounded by

$$\binom{m}{\log m}(m^{-2}\log m)^{\log m} \leq m^{\log m}(m^{-2}\log m)^{\log m} = m^{-\omega(1)}. \tag{142}$$

Taking the union bound over all $m^6 - 1$ possible pairs $(c_1, c_2)$ gives the desired result. $\qquad\square$

We now break down the desired statement (113) into the upper bound and lower bound separately:

$$\overline{h}(c) \leq (3c - 2)(\sqrt{m}\log^2 m + \sigma\sqrt{q}\log m) \tag{143}$$

$$\overline{h}(c) \geq \left(-\frac{3}{c} + 2\right)(\sqrt{m}\log^2 m + \sigma\sqrt{q}\log m). \tag{144}$$

Note that under the high-probability event $|v_i^\nu| \leq \sqrt{m}\log m$ (for $\nu \in \{a, b\}$ and all $i$), we have for all $c$ that

$$|\overline{h}(c)| \overset{(111)}{\leq} \frac{1}{1+c}(|\overline{e}'_a(c)| + c|\overline{e}'_b(c)|) \leq |\overline{e}'_a(c)| + |\overline{e}'_b(c)| \overset{(108)}{\leq} \sum_i |v_i^a| + \sum_i |v_i^b| \leq 2m^{3/2}\log m. \tag{145}$$

Similar to the proof of Lemma 9, we first look at the extreme values of $c$. When $c = m^3$, we have $\overline{h}(c) \leq 2m^{3/2}\log m \leq (3c - 2)(\sqrt{m}\log^2 m + \sigma\sqrt{q}\log m)$ so that (143) holds. When $c = \frac{1}{m^3}$, we have $\overline{h}(c) \geq -2m^{3/2}\log m \geq (-\frac{3}{c} + 2)(\sqrt{m}\log^2 m + \sigma\sqrt{q}\log m)$, where we simply bound $\sigma\sqrt{q} \geq 0$, so that (144) holds.

Suppose by contradiction that (143) and (144) never hold true at the same time. Let $c_2$ be the smallest $c \in C$ such that (143) holds (since $c = m^3$ satisfies (143), such a $c$ exists and the smallest is well defined). Note that $c_2 \neq \frac{1}{m^3}$, since (144) holds when $c = \frac{1}{m^3}$. Let $c_1 = c_2 - \frac{1}{m^3} \in C$. By the definition of $c_2$ and what we assumed (for contradiction), (144) fails for $c_2$ and (143) fails for $c_1 = c_2 - \frac{1}{m^3}$. We will bound $\overline{h}(c_2) - \overline{h}(c_1)$ in two ways and show that this is impossible.

Since (144) fails for $c_2$ and (143) fails for $c_1$, we have

$$\overline{h}(c_2) - \overline{h}(c_1) \leq \left(-\frac{3}{c_2} + 2\right)(\sqrt{m}\log^2 m + \sigma\sqrt{q}\log m) - (3c_1 - 2)(\sqrt{m}\log^2 m + \sigma\sqrt{q}\log m) \tag{146}$$

$$= \left(-\frac{3}{c_2} + 2\right)(\sqrt{m}\log^2 m + \sigma\sqrt{q}\log m) - \left(3c_2 - \frac{3}{m^3} - 2\right)(\sqrt{m}\log^2 m + \sigma\sqrt{q}\log m) \tag{147}$$

$$\leq -(2 - o(1))(\sqrt{m}\log^2 m + \sigma\sqrt{q}\log m), \tag{148}$$

where we used $x + \frac{1}{x} \geq 2$ and $\frac{3}{m^3} = o(1)$.

On the other hand, we have from the definition of $\overline{h}$ in (111) that

$$\overline{h}(c_2) - \overline{h}(c_1) = \frac{1}{1+c_2}(\overline{e}'_a(c_2) - c_2\overline{e}'_b(c_2)) - \frac{1}{1+c_1}(\overline{e}'_a(c_1) - c_1\overline{e}'_b(c_1)) \tag{149}$$

$$= \left(\frac{1}{1+c_2}(\overline{e}'_a(c_2) - c_2\overline{e}'_b(c_2)) - \frac{1}{1+c_1}(\overline{e}'_a(c_2) - c_1\overline{e}'_b(c_2))\right)$$

$$+ \left(\frac{1}{1+c_1}(\overline{e}'_a(c_2) - c_1\overline{e}'_b(c_2)) - \frac{1}{1+c_1}(\overline{e}'_a(c_1) - c_1\overline{e}'_b(c_1))\right) \tag{150}$$

$$= \frac{(c_1 - c_2)(\overline{e}'_a(c_2) + \overline{e}'_b(c_2))}{(1+c_1)(1+c_2)} + \frac{1}{1+c_1}\left[\overline{e}'_a(c_2) - \overline{e}'_a(c_1) - c_1(\overline{e}'_b(c_2) - \overline{e}'_b(c_1))\right]. \tag{151}$$

We proceed to handle the two terms on the right-hand side:

- For the second term, under the high-probability events (i) at most $\log m$ items have $x_i(c_1) \neq x_i(c_2)$ (Lemma 21), and (ii) $|v_i^a| \leq \sqrt{m}\log m$ and $|v_i^b| \leq \sqrt{m}\log m$ (established in (140)), we have

$$|\overline{e}'_a(c_2) - \overline{e}'_a(c_1) - c_1(\overline{e}'_b(c_2) - \overline{e}'_b(c_1))| \leq (1+c_1)\sqrt{m}\log^2 m, \tag{152}$$

and this $1 + c_1$ term cancels with $\frac{1}{1+c_1}$ in (151).

- For the first term, we use the same reasoning as (145) to obtain

$$|\bar{e}'_a(c_2) + \bar{e}'_b(c_2)| \leq 2m^{3/2} \log m, \tag{153}$$

which in turn implies (via $|c_1 - c_2| = \frac{1}{m^3}$) that

$$\frac{(c_1 - c_2)(\bar{e}'_a(c_2) + \bar{e}'_b(c_2))}{(1 + c_1)(1 + c_2)} \leq \frac{1}{m^3} \cdot 2m^{3/2} \log^2 m \leq \frac{1}{m}. \tag{154}$$

Substituting these findings into (151), we have with probability $1 - o(1)$ that

$$|\bar{h}(c_2) - \bar{h}(c_1)| \leq \frac{1}{m} + \sqrt{m} \log^2 m = (1 + o(1))\sqrt{m} \log^2 m. \tag{155}$$

Equations (148) and (155) directly contradict one another, so our assumption that no such $c \in C$ exists must be false. Therefore, there must exist some $c \in C$ such that both (143) and (144) hold.

## D. Proof of Theorem 3 (Lower Bound)

### D.1. Main Proof Steps via Auxiliary Results

We consider the randomized instance specified in Table 1, where the "latent" binary variables $X_1, \ldots, X_m$ are independently drawn from Bernoulli$(1/2)$. We assume for convenience that $m$ is a multiple of 4; if this is not the case, it suffices to round down to the nearest multiple of 4, with any "remainder" items having zero utility for both agents. While the algorithm (namely, its querying strategy and/or allocation rule) may be randomized in general, we have for any notion of "error" (e.g., positive envy) that

$$\mathbb{P}[\text{error}] = \mathbb{E}_{\mathsf{Alg}}[\mathbb{P}[\text{error} \mid \mathsf{Alg}]] \geq \min_{\mathsf{Alg}} \mathbb{P}[\text{error} \mid \mathsf{Alg}], \tag{156}$$

and as a result, in order to establish a lower bound, it suffices to consider deterministic strategies. Note that this reduction to deterministic strategies is standard, in particular being a component of Yao's minimax principle (e.g., see (Motwani & Raghavan, 2010, Sec. 2.2.2)).

We will show that under the conditions of Theorem 3, any deterministic algorithm fails to achieve an envy-free allocation with a certain probability. We will first show in Lemma 22 that with a suitable choice of $\varepsilon$ (for the instance in Table 1), the optimal allocation of this hard instance has envy at most $-\Delta$ with high probability. Then, Lemma 23 introduces two sufficient conditions of any output allocation having positive envy. Lemma 24 and Lemma 25 each provide a query complexity bound corresponding to one of the failing conditions. Then, we will state Theorem 7 (a more precise version of Theorem 3 with $\sigma$ dependence), which combines the two conditions and states the final query complexity bound.

**Lemma 22.** *For any $\delta \in (0, 1)$, $\Delta \in (0, m)$, and $\gamma \in (0, 0.5]$ satisfying $2\sqrt{\log(2/\delta) \cdot m} \leq \frac{m}{2}$ and $\frac{4(\Delta+1)}{m} \leq \frac{1}{2}$, under the choice $\varepsilon = \frac{2(\Delta+1)}{m - 2\sqrt{\log(2/\delta) \cdot m}}$, it holds with probability at least $1 - \delta$ that $\mathrm{OptEnvy} \leq -\Delta$.*

The proof is given in Section D.2. The main idea is to show that there exists an allocation with negative envy at least $\Delta$, which implies that the optimal allocation must also have negative envy at least $\Delta$.

For any output allocation $\widehat{\mathcal{A}}$ based on the noisy queries, we define $\widehat{X} \in \{0, 1\}^m$, where $\widehat{X}_i = 1$ if $i \in \widehat{\mathcal{A}}_a$ and $\widehat{X}_i = 0$ if $i \in \widehat{\mathcal{A}}_b$. Considering any (possibly adaptive) deterministic algorithm that produces $\widehat{\mathcal{A}}$ (or equivalently $\widehat{X}$), we define the following index sets:

$$\widehat{A}_\varepsilon = \left\{i : i \leq \frac{m}{2} \wedge \widehat{X}_i = 1\right\} \tag{157}$$

$$\widehat{B}_\varepsilon = \left\{i : i \leq \frac{m}{2} \wedge \widehat{X}_i = 0\right\}, \tag{158}$$

where the subscript is used to highlight that these items have valuations $\frac{1}{2} \pm \varepsilon$. Moreover, we define $S_i = 1$ if $i$ is allocated to Agent $a$, and $S_i = -1$ otherwise, and let

$$V_\gamma = \sum_{i \,:\, i > m/2} S_i u_i^a. \tag{159}$$

In addition, we let $d_H(X, Y) = |\{i : X_i \neq Y_i\}|$ denote the Hamming distance between two binary vectors of the same length. Then, the following lemma presents sufficient conditions for an allocation to have positive envy. The first condition (160) essentially states that the estimation of the first-half items is "not good enough", and the second condition (161) essentially states that the number of first-half items and/or the utilities of second-half items in the allocation are "too imbalanced".

**Lemma 23.** *For any allocation $\widehat{\mathcal{A}}$, if there exists a constant $K > 0$ such that*

$$d_H(X_{[1:m/2]}, \widehat{X}_{[1:m/2]}) > \frac{m}{4} - \frac{K\gamma\sqrt{m}}{2\varepsilon} \tag{160}$$

*and*

$$\left| \frac{1}{2}(|\widehat{A}_\varepsilon| - |\widehat{B}_\varepsilon|) + V_\gamma \right| \geq K\gamma\sqrt{m}, \tag{161}$$

*then it holds that* $\mathrm{Envy}(\widehat{\mathcal{A}}) > 0$.

The proof is given in Section D.3. The main idea is to express the envy of each agent in terms of $\widehat{A}_\varepsilon, \widehat{B}_\varepsilon, V_\gamma$, and $d_H(X_{[1:m/2]}, \widehat{X}_{[1:m/2]})$. When both conditions hold, the maximal envy between two agents must be positive.

With $q$ denoting the total number of queries, the next lemma presents a threshold for $q$ under which (160) holds with high probability. Before stating the lemma, we highlight that there are two sources of randomness in our setup:

- The latent variables $X_1, \ldots, X_m$ are i.i.d. Bernoulli$(1/2)$ and dictate the items' valuations as summarized in Table 1;

- Once the instance is generated, the algorithm performs $q$ sequentially-chosen queries whose outcomes are random due to the noise. (Recall, however, that we assumed that the algorithm itself is deterministic.)

Accordingly, we let $\mathcal{D}$ denote the "data" collected throughout the course of the algorithm, i.e., the $q$ queries made and their resulting outcomes. While it is most natural to think of generating $\mathbf{X} = (X_1, \ldots, X_m)$ and then generating $\mathcal{D}$ given $\mathbf{X}$, the bulk of our analysis will actually concern the posterior distribution of $\mathbf{X}$ given $\mathcal{D}$.

**Lemma 24.** *For any $c > 0$ and $K > 0$,[11] and for any estimate $\widehat{X}_{[1:m/2]}$, if*

$$\frac{cK\gamma\sqrt{m}}{\varepsilon} \leq 10^{-5} m \tag{162}$$

*and the number of queries satisfies*

$$q \leq \max\left\{ \frac{\sigma cK\gamma\sqrt{m}}{\sqrt{2}\varepsilon^2}, \frac{\sigma^2 c^2 K^2 \gamma^2}{\varepsilon^4} \right\}, \tag{163}$$

*then it holds with probability at least $0.99$ (with respect to $\mathcal{D}$) that both of the following are true: (i) We have*

$$\mathbb{E}\left[ d_H(X_{[1:m/2]}, \widehat{X}_{[1:m/2]}) \,\Big|\, \mathcal{D} \right] \geq \frac{m}{4} - \frac{100cK\gamma\sqrt{m}}{\varepsilon}, \tag{164}$$

*and (ii) there exists a constant $C > 0$ (not depending on $m$ or $\mathcal{D}$) such that*

$$\mathrm{Var}\left[ d_H(X_{[1:m/2]}, \widehat{X}_{[1:m/2]}) \,\Big|\, \mathcal{D} \right] \geq Cm. \tag{165}$$

The proof is given in Section D.4. The main idea is to use Assouad's lemma (Lemma 4 in Appendix A) to derive a lower bound on the Hamming distance. An average Hamming distance of $m/4$ can be achieved by random guessing, and (164) dictates that the average distance is very close to this trivial value. This, in turn, suggests that we have high uncertainty about the value of $X_i$ for most items, which suggests a high variance in estimation accuracy as formalized in (165). Note

---

[11]Here we could "merge" $c$ and $K$ into a single constant replacing $cK$, but the form we present here will be convenient so that we can let $K$ coincide with that in Lemma 23.

also that the reason for $q$ being a maximum of two terms is that our analysis upper bounds the number of queries made to items in $[1:m/2]$, and we take the tighter of two such upper bounds, namely $\min\{q, m/2\}$ (respectively corresponding to the two terms in (163)).

Next, the following lemma presents a threshold for $q$ to make (161) hold. Note that here and throughout the analysis, we made no attempt to optimize constants, since our focus in this paper is on scaling laws.

**Lemma 25.** *For sufficiently large $m$, any $\rho \in (0, 1)$, and any output allocation $\widehat{\mathcal{A}}$, there exists some constant $K$ depending on $\rho$ such that if the number of queries satisfies*

$$q \leq \max\left\{ \frac{10^{-4}\sigma m}{\sqrt{2} \cdot \gamma}, \frac{10^{-8}\sigma^2 m}{\gamma^2} \right\},$$

*then it holds with probability at least $0.99$ (with respect to $\mathcal{D}$) that*

$$\mathbb{P}\left[ \left| \frac{1}{2}(|\widehat{A}_\varepsilon| - |\widehat{B}_\varepsilon|) + V_\gamma \right| \geq K\gamma\sqrt{m} \, \middle| \, \mathcal{D} \right] \geq 1 - \rho.$$

The proof is given in Section D.5 and uses broadly similar ideas to those in the proof of Lemma 24, but with more focus on the second half of the items. In particular, we show that conditioned on $\mathcal{D}$, the quantity $V_\gamma$ can be expressed as an independent sum with sufficient variance to apply a central limit theorem argument, capturing that the fluctuations in $V_\gamma$ are too significant for $\frac{1}{2}(|\widehat{A}_\varepsilon| - |\widehat{B}_\varepsilon|) + V_\gamma$ to be contained in a window of length $2K\gamma\sqrt{m}$ with probability exceeding $\rho$.

By combining the thresholds for both conditions (from Lemma 24 and Lemma 25), the following theorem establishes a threshold on $q$ below which positive envy is unavoidable; specializing this result to the regime $\sigma^2 = \Theta(1)$ recovers Theorem 3.

**Theorem 7.** *For any $\Delta \in (1, m/2)$, there exist constants $c_1 > 0$ and $c_2 > 0$ such that, for any (possibly adaptive and/or randomized) algorithm, if the number of queries satisfies*[12]

$$q \leq \begin{cases} \frac{c_1 \sigma m^{2.5}}{\Delta^2} & \text{when } \Delta = \omega(m^{3/4}), \\ \frac{c_2 \sigma^2 m^{2.5}}{\Delta^2} & \text{when } \Delta = O(m^{3/4}), \end{cases} \tag{166}$$

*then the output allocation $\widehat{\mathcal{A}}$ satisfies the following for sufficiently large $m$:*

$$\mathbb{P}[\text{Envy}(\widehat{\mathcal{A}}) > 0 \wedge \text{OptEnvy} \leq -\Delta] \geq 1/3.$$

The proof is given in Section D.6, and is based on carefully choosing $\rho, K, c$, and $\gamma$ to satisfy both Lemma 24 and Lemma 25, and then using those lemmas to lower bound the probability of having positive envy. In both cases defining $q$, the lower bound has a scaling of $\widetilde{\Omega}(m^{5/2}/\Delta^2)$ as $m \to \infty$ for any constant noise level $\sigma > 0$.

### D.2. Proof of Lemma 22 (Relation Between $\varepsilon$ and $\Delta$)

Define $A_\varepsilon^* = \{i : i \leq \frac{m}{2} \wedge X_i = 1\}$ and $B_\varepsilon^* = \{i : i \leq \frac{m}{2} \wedge X_i = 0\}$, where the subscript highlights that these concern the items with valuations $\frac{1}{2} \pm \varepsilon$. Since $|A_\varepsilon^*| \sim \text{Binomial}(\frac{m}{2}, \frac{1}{2})$, Hoeffding's inequality implies that for any $\delta \in (0, 1)$,

$$\mathbb{P}\left[ \left| |A_\varepsilon^*| - \frac{m}{4} \right| \leq \sqrt{\frac{\log(2/\delta)}{4} \cdot m} \right] \geq 1 - \delta, \tag{167}$$

and since $|B_\varepsilon^*| = \frac{m}{2} - |A_\varepsilon^*|$, we find that this concentration condition on $|A_\varepsilon^*|$ implies the same for $|B_\varepsilon^*|$.

We do not seek to study the exact optimal allocation, but instead construct one that is good enough to establish the lemma:

- When there are at least as many ones as zeros in $X_{[1:\frac{m}{2}]}$, let $A_\varepsilon$ be any set of $m/4$ indices with $X_i = 1$ and let $B_\varepsilon$ be the set of remaining indices from $X_{[1:\frac{m}{2}]}$.

---

[12]Strictly speaking the cases $\omega(m^{3/4})$ and $O(m^{3/4})$ are not exhaustive, but the proof remains unchanged when we change these cases to being above or below $Cm^{3/4}$ for sufficiently large $C$.

- When $X_{[1:\frac{m}{2}]}$ has more zeros than ones, let $B_\varepsilon$ be any set of $m/4$ indices with $X_i = 0$ and let $A_\varepsilon$ be the set of remaining indices from $X_{[1:\frac{m}{2}]}$.

- For $X_{[\frac{m}{2}+1:m]}$, *evenly* divide the ones into $(A_\gamma^1, B_\gamma^1)$ and *evenly* divide the zeros into $(A_\gamma^0, B_\gamma^0)$. If the number of ones is odd, we assign the remainder item to $A_\gamma^1$, and if the number of zeros is odd, we assign the remainder item to $B_\gamma^0$.

Recall the valuations in Table 1. Using those, we observe that the amount by which Agent $a$ envies Agent $b$ due to the allocations in the first dot point is

$$\underbrace{-\frac{m}{4}\left(\frac{1}{2}+\varepsilon\right)}_{a \text{ gets } m/4 \text{ ones}} + \underbrace{|B_\varepsilon^*|\left(\frac{1}{2}-\varepsilon\right)}_{b \text{ gets } |B_\varepsilon^*| \text{ zeros}} + \underbrace{\left(\frac{m}{4}-|B_\varepsilon^*|\right)\left(\frac{1}{2}+\varepsilon\right)}_{b \text{ gets } \frac{m}{4}-|B_\varepsilon^*| \text{ ones}} = -2\varepsilon|B_\varepsilon^*|. \tag{168}$$

Likewise, the amount by which Agent $b$ envies Agent $a$ due to the first dot point is

$$\underbrace{-|B_\varepsilon^*|\left(\frac{1}{2}+\varepsilon\right)}_{b \text{ gets } |B_\varepsilon^*| \text{ zeros}} - \underbrace{\left(\frac{m}{4}-|B_\varepsilon^*|\right)\left(\frac{1}{2}-\varepsilon\right)}_{b \text{ gets } \frac{m}{4}-|B_\varepsilon^*| \text{ ones}} + \underbrace{\frac{m}{4}\left(\frac{1}{2}-\varepsilon\right)}_{a \text{ gets } m/4 \text{ ones}} = -2\varepsilon|B_\varepsilon^*|. \tag{169}$$

Note that the third dot point contributes at most $\frac{1}{2}+\gamma$ envy in either direction. Therefore, when the first dot point holds, we have $-\text{Envy}(\mathcal{A}^*) \geq 2\varepsilon|B_\varepsilon^*| - (\frac{1}{2}+\gamma)$. By the same reasoning, when the second dot point holds, we have $-\text{Envy}(\mathcal{A}^*) \geq 2\varepsilon|A_\varepsilon^*| - (\frac{1}{2}+\gamma)$. Combining these findings gives

$$-\text{Envy}(\mathcal{A}^*) \geq \min\{2\varepsilon|B_\varepsilon^*|, 2\varepsilon|A_\varepsilon^*|\} - \left(\frac{1}{2}+\gamma\right) \tag{170}$$

$$\geq 2\varepsilon \min\{|A_\varepsilon^*|, |B_\varepsilon^*|\} - 1 \tag{171}$$

$$\geq 2\varepsilon\left(\frac{m}{4} - \sqrt{\frac{\log(2/\delta)}{4} \cdot m}\right) - 1 \tag{172}$$

$$= \Delta, \tag{173}$$

where (172) holds when $\left||A_\varepsilon^*| - \frac{m}{4}\right| \leq \sqrt{\frac{\log(2/\delta)}{4} \cdot m}$ and similarly for $B_\varepsilon^*$ (this is true with probability at least $1 - \delta$ by (167)), and (173) follows by setting $\varepsilon = \frac{2(\Delta+1)}{m-2\sqrt{\log(2/\delta)\cdot m}}$. Note that the assumptions $2\sqrt{\log(2/\delta)\cdot m} \leq \frac{m}{2}$ and $\frac{4(\Delta+1)}{m} \leq \frac{1}{2}$ ensure that $\varepsilon \in (0, 1/2]$ and is thus a valid choice.

Hence, with probability at least $1 - \delta$, we have $\text{OptEnvy} \leq \text{Envy}(\mathcal{A}^*) \leq -\Delta$.

### D.3. Proof of Lemma 23 (Sufficient Conditions for Positive Envy)

Recall that $X_{[1:m]}$ are the binary variables defining the instance (see Table 1), $\widehat{X}_{[1:m]}$ are the corresponding "estimates" based on the allocation, and $\widehat{A}_\varepsilon$, $\widehat{B}_\varepsilon$, and $V_\gamma$ are defined in (157)–(159). We define the following useful quantities:

$$N_{aa} = |\{i \leq m/2 : X_i = 1 \wedge \widehat{X}_i = 1\}|, \quad N_{ab} = |\{i \leq m/2 : X_i = 1 \wedge \widehat{X}_i = 0\}| \tag{174}$$

$$N_{ba} = |\{i \leq m/2 : X_i = 0 \wedge \widehat{X}_i = 1\}|, \quad N_{bb} = |\{i \leq m/2 : X_i = 0 \wedge \widehat{X}_i = 0\}|. \tag{175}$$

In other words, $N_{\nu\nu'}$ is the number of items in $\{1, \ldots, m/2\}$ preferred by agent $\nu$ and assigned to agent $\nu'$.

We then have

$$-\text{Envy}_{a \to b}(\widehat{\mathcal{A}}) = \left(\frac{1}{2}+\varepsilon\right)(N_{aa} - N_{ab}) + \left(\frac{1}{2}-\varepsilon\right)(N_{ba} - N_{bb}) + V_\gamma \tag{176}$$

$$= \frac{1}{2}(|\widehat{A}_\varepsilon| - |\widehat{B}_\varepsilon|) + \varepsilon(N_{aa} + N_{bb} - N_{ab} - N_{ba}) + V_\gamma \tag{177}$$

$$= \frac{1}{2}(|\widehat{A}_\varepsilon| - |\widehat{B}_\varepsilon|) + \varepsilon\left(\frac{m}{2} - 2d_H(X_{[1:m/2]}, \widehat{X}_{[1:m/2]})\right) + V_\gamma, \tag{178}$$

$$-\text{Envy}_{b\to a}(\widehat{\mathcal{A}}) = \left(\frac{1}{2}+\varepsilon\right)(N_{bb}-N_{ba}) + \left(\frac{1}{2}-\varepsilon\right)(N_{ab}-N_{aa}) - V_{\gamma} \tag{179}$$

$$= \frac{1}{2}(|\widehat{B}_{\varepsilon}|-|\widehat{A}_{\varepsilon}|) + \varepsilon(N_{aa}+N_{bb}-N_{ab}-N_{ba}) - V_{\gamma} \tag{180}$$

$$= \frac{1}{2}(|\widehat{B}_{\varepsilon}|-|\widehat{A}_{\varepsilon}|) + \varepsilon\left(\frac{m}{2}-2d_H(X_{[1:m/2]},\widehat{X}_{[1:m/2]})\right) - V_{\gamma}, \tag{181}$$

where:

- (177) follows since $N_{aa}+N_{ba} = |\widehat{A}_{\varepsilon}|$ and $N_{ab}+N_{bb}=|\widehat{B}_{\varepsilon}|$;

- (178) follows since $N_{aa}+N_{ab}+N_{ba}+N_{bb} = \frac{m}{2}$ and $N_{ab}+N_{ba} = d_H(X_{[1:m/2]},\widehat{X}_{[1:m/2]})$;

- The steps for $-\text{Envy}_{b\to a}(\widehat{\mathcal{A}})$ giving (181) are entirely analogous.

When $\left|\frac{1}{2}(|\widehat{A}_{\varepsilon}|-|\widehat{B}_{\varepsilon}|)+V_{\gamma}\right| \geq K\gamma\sqrt{m}$ for some constant $K$ and $d_H(X_{[1:m/2]},\widehat{X}_{[1:m/2]}) > \frac{m}{4} - \frac{K\gamma\sqrt{m}}{2\varepsilon}$, the overall envy is

$$-\text{Envy}(\widehat{\mathcal{A}}) = \min\{-\text{Envy}_{a\to b}(\widehat{\mathcal{A}}), -\text{Envy}_{b\to a}(\widehat{\mathcal{A}})\} \tag{182}$$

$$= -\left|\frac{1}{2}(|\widehat{A}_{\varepsilon}|-|\widehat{B}_{\varepsilon}|)+V_{\gamma}\right| + \varepsilon\left(\frac{m}{2}-2d_H(X_{[1:m/2]},\widehat{X}_{[1:m/2]})\right) \tag{183}$$

$$< -K\gamma\sqrt{m} + K\gamma\sqrt{m} \tag{184}$$

$$= 0, \tag{185}$$

where the second line follows by combining (178) and (181).

### D.4. Proof of Lemma 24 (Lower Bound on Hamming Distance)

To reduce notation, we define $Z = X_{[1:m/2]}$ and $\widehat{Z} = \widehat{X}_{[1:m/2]}$. Recall that $q$ is the total number of queries, and let $Q_i$ denote the (random) number of queries to item $i$. We proceed as follows:

$$\mathbb{E}[d_H(Z,\widehat{Z})] \geq \frac{1}{2}\sum_{i=1}^{m/2}\left(1-\|P_i^+ - P_i^-\|_{\text{TV}}\right) \tag{186}$$

$$\geq \frac{1}{2}\sum_{i=1}^{m/2}\left(1-\sqrt{\frac{1}{2}D_{\text{KL}}(P_i^+\|P_i^-)}\right) \tag{187}$$

$$\geq \frac{m}{4} - \sum_{i\leq m/2}\sqrt{\frac{2\mathbb{E}_{P_i^+}[Q_i]\varepsilon^2}{\sigma^2}} \tag{188}$$

$$= \frac{m}{4} - \frac{\varepsilon\sqrt{2}}{\sigma}\sum_{i\,:\,i\leq m/2\,\wedge\,\mathbb{E}_{P_i^+}[Q_i]>0}\sqrt{\mathbb{E}_{P_i^+}[Q_i]} \tag{189}$$

$$\geq \frac{m}{4} - \frac{\varepsilon\sqrt{2}}{\sigma}\sqrt{q\cdot|\{i:i\leq m/2\,\wedge\,\mathbb{E}_{P_i^+}[Q_i]>0\}|} \tag{190}$$

$$\geq \frac{m}{4} - \frac{\varepsilon\sqrt{2}}{\sigma}\sqrt{q\min\left\{q,\frac{m}{2}\right\}} \tag{191}$$

where:

- (186) applies Assouad's method (see Lemma 4 in Appendix A) where $P_i^+$ and $P_i^-$ are the distributions on the query outcome sequence[13] $(Y_1,\ldots,Y_q)$ given $X_i = 1$ and $X_i = 0$ respectively; recall that each element of $Z$ is drawn from Bernoulli$(\frac{1}{2})$.

---

[13]This sequence also determines the sequence of queries made itself, because we have assumed a deterministic algorithm.

- (187) follows from Pinsker's inequality.

- We show (188) in several sub-steps:

  - The KL divergence between two Gaussians with means $\mu, \mu'$ and variance $\sigma^2$ is well known to be $\frac{(\mu-\mu')^2}{2\sigma^2}$, which implies

  $$D_{\mathrm{KL}}\left(N\left(\frac{1}{2}+\varepsilon, \sigma^2\right) \,\middle\|\, N\left(\frac{1}{2}-\varepsilon, \sigma^2\right)\right) = \frac{2\varepsilon^2}{\sigma^2}, \tag{192}$$

  and similarly when we swap $\frac{1}{2}+\varepsilon$ and $\frac{1}{2}-\varepsilon$ with one another.

  - Hence, the KL divergence between the outcome distributions of a given item $i \in [1 : m/2]$ with $X_i = 1$ vs. $X_i = 0$ is $\frac{4\varepsilon^2}{\sigma^2}$, where the factor of 2 is doubled to 4 because each query consists of two observed values, one per agent (the two are independent, and KL divergence is additive for independent product distributions).

  - Finally, we obtain $D_{\mathrm{KL}}(P_i^+ \| P_i^-) \le \mathbb{E}_{P_i^+}[Q_i] \cdot \frac{4\varepsilon^2}{\sigma^2}$ via a standard argument based on the chain rule for KL divergence; for completeness, we provide the details as follows. Recall that the utilities for items in $[1 : m/2]$ are specified by the binary variables $X_1, \ldots, X_{m/2}$ as per Table 1, and let $X_{(-i)}$ be the set of such variables excluding $X_i$. By definition, we have for any outcome sequence $\mathbf{y} = (y_1, \ldots, y_q)$ that $P_i^+(\mathbf{y}) = \mathbb{E}_{X_{(-i)}}\big[P(\mathbf{y} \,|\, X_i = 1, X_{(-i)})\big]$ and $P_i^-(\mathbf{y}) = \mathbb{E}_{X_{(-i)}}\big[P(\mathbf{y} \,|\, X_i = 0, X_{(-i)})\big]$, where $P(\mathbf{y} \,|\, \ldots)$ denotes the conditional probability of observing $\mathbf{y}$ given the specified $X$ values. In other words, $P_i^+$ and $P_i^-$ are mixture distributions over $X_{(-i)}$. As a result, the convexity of KL divergence (Lemma 5) gives

  $$D_{\mathrm{KL}}(P_i^+ \| P_i^-) \le \mathbb{E}_{X_{(-i)}}\big[D_{\mathrm{KL}}\big(P(\cdot \,|\, X_i = 1, X_{(-i)}) \,\|\, P(\cdot \,|\, X_i = 0, X_{(-i)})\big)\big]. \tag{193}$$

  The two distributions in the arguments to $D_{\mathrm{KL}}$ only differ in the distribution of the $i$-th item, meaning that we can apply the form of the chain rule in (16) in Lemma 6 to obtain

  $$D_{\mathrm{KL}}(P_i^+ \| P_i^-) \le \mathbb{E}_{X_{(-i)}}\left[\mathbb{E}[Q_i \,|\, X_i = 1, X_{(-i)}] \cdot \frac{4\varepsilon^2}{\sigma^2}\right], \tag{194}$$

  where the inner expectation corresponds to the expectation in (16) with $P(\cdot | X_i = 1, X_{(-i)})$ in place of $P$ and $Q_i$ in place of $N_i$, and the term $\frac{4\varepsilon^2}{\sigma^2}$ comes from the KL divergence calculation in the previous dot point. By the law of total expectation, (194) simplifies to $\mathbb{E}_{P_i^+}[Q_i] \cdot \frac{4\varepsilon^2}{\sigma^2}$, as claimed.

- (190) follows by Jensen's inequality applied to the uniform distribution on $\left|\{i : i \le m/2 \wedge \mathbb{E}_{P_i^+}[Q_i] > 0\}\right|$ and the fact that $\sum_{i \le m/2} \mathbb{E}_{P_i^+}[Q_i] \le q$, which in turn holds because $\sum_i Q_i \le q$.

- (191) follows from $\left|\{i : i \le m/2 \wedge \mathbb{E}_{P_i^+}[Q_i] > 0\}\right| \le \min\{q, m/2\}$, for which the upper bound of $m/2$ is trivial, so it remains to show an upper bound of $q$ whenever $q < m/2$. To see this, we claim that due to the i.i.d. prior on $(X_1, \ldots, X_{m/2})$, if an algorithm makes $q < \frac{m}{2}$ queries then it can be assumed without loss of generality that it never queries items in $[q+1 : m/2]$. This is because ruling out such items still leaves the remaining items $[1 : q]$ that can all be queried, and having $q$ such items is the highest number possible because the query budget is only $q$. (A different subset of $q$ items in $[1 : m/2]$ could be queried, but the above choice is without loss of generality since the i.i.d. prior on $X_{[1:m/2]}$ is invariant to re-ordering.) Recall also that we have reduced to the case of deterministic querying strategies as explained following (156); this rules out the possibility of having more positive values of $\mathbb{E}_{P_i^+}[Q_i]$ due to randomization.

When $q \le \max\left\{\frac{\sigma c K \gamma \sqrt{m}}{\sqrt{2} \cdot \varepsilon^2}, \frac{\sigma^2 c^2 K^2 \gamma^2}{\varepsilon^4}\right\}$ as assumed in (163), we obtain from (191) that

$$\mathbb{E}[d_H(Z, \widehat{Z})] \ge \frac{m}{4} - \frac{\varepsilon\sqrt{2}}{\sigma}\sqrt{q \min\{q, m/2\}} \tag{195}$$

$$= \frac{m}{4} - \min\left\{\frac{q\varepsilon\sqrt{2}}{\sigma}, \frac{\varepsilon\sqrt{qm}}{\sigma}\right\} \tag{196}$$

$$\geq \frac{m}{4} - \frac{cK\gamma\sqrt{m}}{\varepsilon}. \tag{197}$$

Now, given any algorithm that produced $\widehat{Z}$, let $\widehat{Z}'$ denote the following alternative output:

$$\widehat{Z}' = \begin{cases} \widehat{Z} & \text{when } \mathbb{E}[d_H(Z, \widehat{Z}) \,|\, \mathcal{D}] \leq \frac{m}{4}, \\ \mathbf{1} - \widehat{Z} & \text{otherwise.} \end{cases} \tag{198}$$

By this definition, we always have $\mathbb{E}[d_H(Z, \widehat{Z}') \,|\, \mathcal{D}] \leq \frac{m}{4}$. Moreover, since (197) applies to an *arbitrary* algorithm output, we can apply it to $\widehat{Z}'$ to obtain $\mathbb{E}[d_H(Z, \widehat{Z}')] \geq \frac{m}{4} - \frac{cK\gamma\sqrt{m}}{\varepsilon}$. Since

$$\mathbb{E}_{\mathcal{D}}\left[\frac{m}{4} - \mathbb{E}[d_H(Z, \widehat{Z}') \,|\, \mathcal{D}]\right] = \frac{m}{4} - \mathbb{E}[d_H(Z, \widehat{Z}')] \leq \frac{cK\gamma\sqrt{m}}{\varepsilon},$$

by Markov's inequality, it holds with probability at most $0.01$ (with respect to $\mathcal{D}$) that

$$\frac{m}{4} - \mathbb{E}[d_H(Z, \widehat{Z}') \,|\, \mathcal{D}] \geq \frac{100cK\gamma\sqrt{m}}{\varepsilon}.$$

Hence, with probability at least $0.99$ (with respect to $\mathcal{D}$),

$$\mathbb{E}[d_H(Z, \widehat{Z}) \,|\, \mathcal{D}] \geq \mathbb{E}[d_H(Z, \widehat{Z}') \,|\, \mathcal{D}] \geq \frac{m}{4} - \frac{100cK\gamma\sqrt{m}}{\varepsilon}, \tag{199}$$

where the first step follows directly from the definition of $\widehat{Z}'$. This completes the proof of (164).

Towards establishing (165), we first use the assumption $\frac{cK\gamma\sqrt{m}}{\varepsilon} \leq 10^{-5}m$ in (162) to further lower bound (199) by

$$\mathbb{E}[d_H(Z, \widehat{Z}) \,|\, \mathcal{D}] \geq 0.249m. \tag{200}$$

In the following analysis, we condition on a specific $\mathcal{D}$ satisfying (200). Define $C_0 = 0.001$, and let $\widetilde{Z}$ be an estimate of $Z$ such that $\widetilde{Z}_i$ is the maximum posterior probability estimate (i.e., the value in $\{0, 1\}$ with the higher probability conditioned on $\mathcal{D}$, breaking ties arbitrarily) when $\mathrm{Var}[Z_i \,|\, \mathcal{D}] \leq C_0(1 - C_0)$, and otherwise $\widetilde{Z}_i$ is independently drawn from Bernoulli$(1/2)$. Let $S$ denote the set of indices $i$ with $\mathrm{Var}[Z_i \,|\, \mathcal{D}] \leq C_0(1 - C_0)$. Defining $\nu_i = \mathbb{P}[Z_i \neq \widetilde{Z}_i \,|\, \mathcal{D}]$, we first show that for each $i \in S$, it must hold that $\nu_i \leq C_0$. To see this, we write

$$\mathrm{Var}[Z_i \,|\, \mathcal{D}] = \mathbb{P}[Z_i = 0 \,|\, \mathcal{D}] \cdot \mathbb{P}[Z_i = 1 \,|\, \mathcal{D}] = \nu_i(1 - \nu_i), \tag{201}$$

with the first step using that Bernoulli$(p)$ has variance $p(1 - p)$, and the second step using that given $\mathcal{D}$, the quantity $\widetilde{Z}_i$ is a deterministic value in $\{0, 1\}$. Since $\mathrm{Var}[Z_i \,|\, \mathcal{D}] \leq C_0(1 - C_0)$, it follows that either $\nu_i \leq C_0$ or $\nu_i \geq 1 - C_0$, and the maximum posterior probability strategy guarantees $\nu_i \leq \frac{1}{2}$. Hence, we must have $\nu_i \leq C_0$ for each $i \in S$.

Next, we show that $|S| < 0.01m$. Supposing for contradiction that $|S| \geq 0.01m$, we have

$$\mathbb{E}[d_H(Z, \widetilde{Z}) \,|\, \mathcal{D}] = \sum_{i=1}^{m/2} \mathbb{P}[Z_i \neq \widetilde{Z}_i \,|\, \mathcal{D}] \tag{202}$$

$$= \sum_{i \in S} \nu_i + \sum_{i \notin S} \nu_i \tag{203}$$

$$\leq |S| \max_{i \in S} \nu_i + \left(\frac{m}{2} - |S|\right) \cdot \frac{1}{2} \tag{204}$$

$$\leq C_0 m + 0.245m \tag{205}$$

$$= 0.246m, \tag{206}$$

with the first inequality again using $\nu_i \leq \frac{1}{2}$. This result contradicts the preceding result on $\mathbb{E}[d_H(Z, \cdot) \,|\, \mathcal{D}]$ in (200). Hence, there must only exist at most $0.01m$ indices with $\mathrm{Var}[Z_i \,|\, \mathcal{D}] \leq C_0(1 - C_0)$, and at least $0.99m$ indices with $\mathrm{Var}[Z_i \,|\, \mathcal{D}] \geq C_0(1 - C_0)$.

We are interested in $\text{Var}[d_H(Z, \widehat{Z}) \,|\, \mathcal{D}]$, and since $d_H(Z, \widehat{Z}) = \sum_{i=1}^{m/2} \mathbf{1}\{Z_i \neq \widehat{Z}_i\}$, one may be concerned with whether the $Z_i$'s are still independent given $\mathcal{D}$. (Note that $\widehat{Z}_i$ is deterministic given $\mathcal{D}$.) Fortunately, this is indeed the case, according to the following known result (which can be proved in a few lines using Bayes' rule).

**Lemma 26.** (Cai et al., 2023, Lemma 5) *Under an i.i.d. prior on $Z_1, \ldots, Z_{m/2}$ and independent noise between queries, conditioned on any collection $\mathcal{D}$ of the $q$ query-outcome pairs, the variables $Z_1, \ldots, Z_{m/2}$ are independent.*

Using this lemma and defining $C = 0.99 C_0 (1 - C_0)$, we deduce that

$$\text{Var}[d_H(Z, \widehat{Z}) \,|\, \mathcal{D}] = \sum_{i=1}^{m/2} \mathbb{P}[Z_i = 1 \,|\, \mathcal{D}] \cdot \mathbb{P}[Z_i = 0 \,|\, \mathcal{D}] \tag{207}$$

$$= \sum_{i=1}^{m/2} \text{Var}[Z_i \,|\, \mathcal{D}] \tag{208}$$

$$\geq 0.99 C_0 (1 - C_0) m \tag{209}$$

$$= Cm, \tag{210}$$

which completes the proof.

### D.5. Proof of Lemma 25 (Lower Bound on $\Theta(\sqrt{m})$ Deviation Probability)

The proof of Lemma 24 is concerned with the first half of the items (see Table 1), but the arguments up to (191) apply verbatim[14] to the second half of the items upon replacing $\varepsilon$ by $\gamma$. Thus, adopting the shorthand $\zeta_\gamma(q, m) = \frac{\gamma\sqrt{2}}{\sigma}\sqrt{q \min\{q, \frac{m}{2}\}}$ for brevity, we have the following analog of (196):

$$\mathbb{E}[d_H(X_{[m/2+1:m]}, \widehat{X}_{[m/2+1:m]})] \geq \frac{m}{4} - \zeta_\gamma(q, m). \tag{211}$$

By considering an "alternative output" in the same way as (198) (with Hamming distance to $X_{[m/2+1:m]}$ at most $m/4$ almost surely) and applying Markov's inequality, we obtain from (211) that the following holds with probability at least $0.99$ (with respect to $\mathcal{D}$), in analogy with (199):

$$\mathbb{E}[d_H(X_{[m/2+1:m]}, \widehat{X}_{[m/2+1:m]}) \,|\, \mathcal{D}] \geq \frac{m}{4} - 100 \zeta_\gamma(q, m). \tag{212}$$

Under the condition $\zeta_\gamma(q, m) \leq 10^{-4} m$ (to be verified shortly), this can further be lower bounded by $0.24m$, thus providing an analog of (200) (with a slightly modified constant). As a result, we can follow identical reasoning to (201)–(210) to conclude that there exists some constant $C'$ such that

$$\text{Var}[d_H(X_{[m/2+1:m]}, \widehat{X}_{[m/2+1:m]}) \,|\, \mathcal{D}] \geq C'm. \tag{213}$$

The above-mentioned condition $\zeta_\gamma(q, m) \leq 10^{-4} m$ can be rewritten as

$$\zeta_\gamma(q, m) = \min\left\{ \frac{\gamma q \sqrt{2}}{\sigma}, \frac{\gamma\sqrt{qm}}{\sigma} \right\} \leq 10^{-4} m, \tag{214}$$

which is equivalent to

$$q \leq \max\left\{ \frac{10^{-4} \sigma m}{\sqrt{2} \cdot \gamma}, \frac{10^{-8} \sigma^2 m}{\gamma^2} \right\} \tag{215}$$

as we have already assumed.

In the lemma statement, we are interested in the quantity $W = \frac{1}{2}(|\widehat{A}_\varepsilon| - |\widehat{B}_\varepsilon|) + V_\gamma$. Since $\widehat{A}_\varepsilon$ and $\widehat{B}_\varepsilon$ are deterministic given $\mathcal{D}$ (due to the algorithm being deterministic), we are interested in understanding the fluctuations introduced by $V_\gamma$,

---

[14]In this part of the analysis, the distinction between an item being "slightly favored by both agents" vs. "slightly favored by one and slightly disfavored by the other" is inconsequential.

which we recall from (159) equals the sum over $i > m/2$ of (true) utilities $u_i^a$ of Agent $a$ weighted by $S_i = 1$ (if allocated to $a$) or $S_i = -1$ (if allocated to $b$). To write $V_\gamma$ in a more explicit form, it is useful to note the following for $i > m/2$:

$$u_i^a = \frac{1}{2} + \gamma(-1 + 2X_i), \tag{216}$$

which follows directly from Table 1 by considering $X_i = 0$ and $X_i = 1$ separately. Multiplying by $S_i$ and summing over $i > m/2$, we deduce that $V_\gamma$ takes the form

$$V_\gamma = 2\gamma \sum_{i\,:\,i>m/2} S_i X_i + c_{\mathcal{D}}, \tag{217}$$

where $c_{\mathcal{D}}$ is deterministic given $\mathcal{D}$ (due to the same being true of the allocation variables $S_i$).

Combining these findings with Lemma 26, we see that $V_\gamma$ given $\mathcal{D}$ is a sum of independent random variables. Moreover, the randomness comes entirely from $\sum_{i\,:\,i>m/2} S_i X_i$, and we observe that when $\mathcal{D}$ satisfies (213), we have

$$\mathrm{Var}\left[\sum_{i\,:\,i>m/2} S_i X_i \,\Big|\, \mathcal{D}\right] = \sum_{i\,:\,i>m/2} \mathrm{Var}[X_i \,|\, \mathcal{D}] \tag{218}$$

$$= \sum_{i\,:\,i>m/2} \mathbb{P}[X_i = 0 \,|\, \mathcal{D}] \cdot \mathbb{P}[X_i = 1 \,|\, \mathcal{D}] \tag{219}$$

$$= \mathrm{Var}[d_H(X_{[m/2+1:m]}, \widehat{X}_{[m/2+1:m]}) \,|\, \mathcal{D}] \tag{220}$$

$$\geq C'm, \tag{221}$$

where (218) uses the above-mentioned independence and $S_i \in \{-1, 1\}$, (219) uses the fact that $\mathrm{Bernoulli}(p)$ has variance $p(1-p)$, (220) is the direct counterpart to (207), and (221) follows from (213).

We now put the above findings together to characterize $W$. We use (217) to write $W = 2\gamma \sum_{i\,:\,i>m/2} S_i X_i + c'_{\mathcal{D}}$ for some $c'_{\mathcal{D}}$ (deterministic given $\mathcal{D}$), and observe that

$$\mathbb{P}[|W| < t \,|\, \mathcal{D}] = \mathbb{P}\left[\left|\sum_{i\,:\,i>m/2} S_i X_i + \frac{c'_{\mathcal{D}}}{2\gamma}\right| < \frac{t}{2\gamma} \,\Big|\, \mathcal{D}\right] \tag{222}$$

$$\leq \frac{t}{\gamma} \cdot \frac{1}{\sqrt{2\pi C'm}} + O\left(\frac{1}{\sqrt{m}}\right), \tag{223}$$

where we applied the Berry–Esseen theorem (Lemma 3 in Appendix A) to $\sum_{i\,:\,i>m/2} S_i X_i$ (with the substitutions $V_T \leftarrow \mathrm{Var}[\sum_{i>m/2} S_i X_i \,|\, \mathcal{D}] \geq C'm$ due to (221), and $\Psi_T \leftarrow O(m)$ due to $S_i X_i \in [-1, 1]$), and used the fact that the resulting Gaussian density is uniformly upper bounded by $\frac{1}{\sqrt{2\pi C'm}}$.[15] Setting $t = \rho\gamma\sqrt{\pi C'm}$, we find that (223) is at most $\frac{\rho}{\sqrt{2}} + O\left(\frac{1}{\sqrt{m}}\right)$, and is therefore at most $\rho$ when $m$ is sufficiently large, thus implying the lemma with $K = \rho\sqrt{\pi C'}$.

### D.6. Completing the Proof of Theorem 7

We first apply the same central limit theorem argument to the items in $[1 : m/2]$ as we did in the preceding steps for the items in $[m/2+1 : m]$. Here we are directly interested in $d_H(X_{[1:m/2]}, \widehat{X}_{[1:m/2]})$, so there is no need for analogs of $S_i$ and $c_{\mathcal{D}}$ used above. The analog of (221) is stated directly in the last part of Lemma 24 (holding with probability at least 0.99), with the constant now denoted as $C$ rather than $C'$.

From these observations, we have with probability at least 0.99 (with respect to $\mathcal{D}$) that the posterior of $d_H(X_{[1:m/2]}, \widehat{X}_{[1:m/2]})$ is asymptotically Gaussian with variance at least $Cm$ as $m \to \infty$. To make this statement more precise, let $T$ follow a Gaussian distribution with the same mean and variance as $d_H(X_{[1:m/2]}, \widehat{X}_{[1:m/2]})$ (and thus

---

[15]We note that $\frac{t}{\gamma} \cdot \frac{1}{\sqrt{2\pi C'm}}$ serves as a *uniform* upper bound for the probability of the Gaussian random variable falling in *any* interval of length $\frac{t}{\gamma}$, so the precise value of $\mathbb{E}[\sum_{i>m/2} S_i X_i \,|\, \mathcal{D}]$ is not needed, and the shift by $\frac{c'_{\mathcal{D}}}{2\gamma}$ in (222) is similarly inconsequential.

variance at least $Cm$). Then, conditioned on any $\mathcal{D}$ satisfying the last part of Lemma 24, the Berry–Esseen theorem (Lemma 3 in Appendix A, with $V_T \leftarrow \Omega(m)$ and $\Psi_T \leftarrow O(m)$ similarly to (223)) gives

$$\mathbb{P}\left[d_H(X_{[1:m/2]}, \widehat{X}_{[1:m/2]}) > \frac{m}{4} - \frac{K\gamma\sqrt{m}}{2\varepsilon}\,\bigg|\,\mathcal{D}\right] \geq \mathbb{P}\left[T > \frac{m}{4} - \frac{K\gamma\sqrt{m}}{2\varepsilon}\right] - O\left(\frac{1}{\sqrt{m}}\right). \tag{224}$$

In the following, we set $\rho = 0.001$ and $c = \frac{1}{200}$, let $K$ be chosen according to the statement of Lemma 25, and let D denote the collection of $\mathcal{D}$'s that simultaneously satisfy Lemma 24 and Lemma 25. The precise choice of $\gamma$ will be specified later, but will be ensured to satisfy $\frac{cK\gamma\sqrt{m}}{\varepsilon} \leq 10^{-5}m$ as required in Lemma 24. Then, we observe that when

$$q \leq \max\left\{\frac{\sigma cK\gamma\sqrt{m}}{\sqrt{2}\cdot\varepsilon^2}, \frac{\sigma^2 c^2 K^2 \gamma^2}{\varepsilon^4}\right\} \tag{225}$$

and

$$q \leq \max\left\{\frac{10^{-4}\sigma m}{\sqrt{2}\cdot\gamma}, \frac{10^{-8}\sigma^2 m}{\gamma^2}\right\}, \tag{226}$$

as assumed in Lemma 24 and Lemma 25 respectively, we have

$$\mathbb{P}[\mathrm{Envy}(\widehat{\mathcal{A}}) > 0] \tag{227}$$

$$\geq \sum_{\mathcal{D}\in\mathsf{D}}\mathbb{P}[\mathcal{D}]\cdot\mathbb{P}\left[\left|\frac{1}{2}(|\widehat{A}_\varepsilon| - |\widehat{B}_\varepsilon|) + V\right| \geq K\gamma\sqrt{m} \,\wedge\, d_H(X_{[1:m/2]}, \widehat{X}_{[1:m/2]}) > \frac{m}{4} - \frac{K\gamma\sqrt{m}}{2\varepsilon}\,\bigg|\,\mathcal{D}\right] \tag{228}$$

$$\geq \sum_{\mathcal{D}\in\mathsf{D}}\mathbb{P}[\mathcal{D}]\left(\mathbb{P}\left[d_H(X_{[1:m/2]}, \widehat{X}_{[1:m/2]}) > \frac{m}{4} - \frac{K\gamma\sqrt{m}}{2\varepsilon}\,\bigg|\,\mathcal{D}\right] - \rho\right) \tag{229}$$

$$\geq \sum_{\mathcal{D}\in\mathsf{D}}\mathbb{P}[\mathcal{D}]\left(\mathbb{P}\left[T > \frac{m}{4} - \frac{K\gamma\sqrt{m}}{2\varepsilon}\right] - o(1) - \rho\right) \tag{230}$$

$$\geq \sum_{\mathcal{D}\in\mathsf{D}}\mathbb{P}[\mathcal{D}]\left(\mathbb{P}\left[T > \mathbb{E}[d_H(X_{[1:m/2]}, \widehat{X}_{[1:m/2]})\,|\,\mathcal{D}]\right] - o(1) - \rho\right) \tag{231}$$

$$= 0.98\cdot\left(\frac{1}{2} - o(1) - 0.001\right) \tag{232}$$

$$= 0.48902 - o(1), \tag{233}$$

where:

- (228) follows from Lemma 23;

- (229) follows from Lemma 25;

- (230) follows from (224);

- (231) follows since Lemma 24 (specifically (164), for which we have already specified $c = \frac{1}{200}$) implies for all $\mathcal{D}\in\mathsf{D}$ that $\mathbb{E}[d_H(X_{[1:m/2]}, \widehat{X}_{[1:m/2]})\,|\,\mathcal{D}] \geq \frac{m}{4} - \frac{K\gamma\sqrt{m}}{2\varepsilon}$;

- (232) follows since a Gaussian exceeds its mean with probability $\frac{1}{2}$, and since we have specified $\rho = 0.001$.

Hence, choosing $\varepsilon$ as in Lemma 22, we have for any $\delta \in (0, 0.489)$ that

$$\mathbb{P}[\mathrm{Envy}(\widehat{\mathcal{A}}) > 0 \wedge \mathrm{OptEnvy} \leq -\Delta] \geq \mathbb{P}[\mathrm{Envy}(\widehat{\mathcal{A}}) > 0] - \delta$$
$$\geq 0.489 - \delta - o(1).$$

Choosing $\delta < 0.15$ ensures that this exceeds $1/3$ for sufficiently large $m$.

Now, the only remaining step is to choose $\gamma$ satisfying (225)–(226) (as well as $\frac{cK\gamma\sqrt{m}}{\varepsilon} \leq 10^{-5}m$). Due to the "max" operations in these equations, it suffices for $q$ to be upper bounded by *either* of the two terms in each one. Recalling from Lemma 22 that $\varepsilon = \Theta(\Delta/m)$ (whenever $\Delta \geq 1$ and thus $1 + \Delta = \Theta(\Delta)$), we have the following:

- When $\Delta = \omega(m^{3/4})$, we set $\gamma = \frac{1}{2}$ and take the first terms in (225) and (226). The term from (225) scales as $\Theta\left(\frac{\sigma m^{2.5}}{\Delta^2}\right)$ and the term from (226) scales as $\Theta(\sigma m)$, and thus the former dominates due to $\Delta = \omega(m^{3/4})$.

- When $\Delta = O(m^{3/4})$, we set $\gamma = c' m^{1/4} \varepsilon$ for some $c' > 0$ sufficiently small to ensure $\gamma \in (0, 1/2)$, and take the second terms in (225) and (226). Both of these terms scale as $\Theta\left(\frac{\sigma^2 m^{2.5}}{\Delta^2}\right)$ under this choice.

Observe that these scalings of $q$ match those stated in (166). Moreover, in both cases, we have $\frac{cK\gamma\sqrt{m}}{\varepsilon} = o(m)$, thus being below $10^{-5}m$ (for sufficiently large $m$) as assumed earlier. This completes the proof.

