# OpenReview forum: "Envy-Free Allocation of Indivisible Goods via Noisy Queries"
_ICML.cc/2026/Conference — ICML 2026 regular_

### Official Review · Reviewer_2nQd · 2026-03-08

**Soundness:** 3
**Presentation:** 4
**Significance:** 3
**Originality:** 3
**Overall Recommendation:** 5
**Confidence:** 3

**Summary:**

In the paper, the authors discuss the classic problem of fair allocations with indivisible goods and valuation queries, and introduce the assumption that the queries might be noisy. The work focuses on the case with additive valuations and two agents, and illustrates that the added element of noisy queries significantly complicates the difficulty of designing and analyzing an algorithm. To help ease the analysis, a variable $\Delta$ is introduced for a given instance to represent the extent to which the two agents do not envy each other in the optimal allocation. It is assumed that its value is at least slightly positive ($\Delta \gg m^\frac{1}{4}$), indicating that an EF allocation always exists. Under this assumption, the authors propose an algorithm that finds an EF allocation with high probability using $O(\frac{M^3}{\Delta^2})$ noisy queries, supported by a lower-bound instance with the same time complexity within logarithmic factors. Expansion of this result to other fairness notions and noise levels is also discussed.

**Compliance With Llm Reviewing Policy:**

Affirmed.

**Final Justification:**

The authors' rebuttal has addressed my questions and reinforced my initial assessment. Although the paper relies on a fairly strong assumption, I believe it explores a promising area and provides a solid foundation for future research. Therefore, I maintain my recommendation to accept.

**Key Questions For Authors:**

1. If I understand it correctly, in Line 890, when $\Delta$ is big, $q$ needs to be large enough for the algorithm to work, which implies $m$ also needs to be large (since the assumption is that $q<m$). So this means the proposed algorithm needs to work for instances with a relatively large number of items (which also makes sense, since brute force is practically sufficient for smaller $m$). Therefore, I wonder if you could provide more examples of the real-life problems to which this algorithm can be applied with these constraints.
2. Throughout the paper, we are using the value $\Delta$ that we are not supposed to know given an instance. Is there any way to estimate its value so we can get a rough estimate of the number of queries required for a possible EF allocation?
3. Assume $\Delta$ is known. Given an instance, we do not know how many queries are needed for the algorithms to produce an EF allocation with high probability. Therefore, I wonder whether we can claim, in the other direction, that for any sufficiently large $q$, the algorithm proposed in Section 4 always gives an EF allocation with higher probability than the simple algorithm in Section 3.

**Limitations:**

yes

**Strengths And Weaknesses:**

## Soundness
The work is technically sound, and the theorems seem to be well supported (I did not thoroughly check the proofs in the Appendix). The authors clearly justified the scope of the setting with convincing arguments.

## Presentation
The article is very well written and structured. It is easy to follow despite having unintuitive notions like $\Delta$. Although there is no formal proof in the main body, a clear, intuitive overview is provided to help readers understand the article's flow. The literature review provides a clear position for the article within related topics.

## Significance
This work opens a new space in the field of Fair division by introducing noise into the valuations of querying agents. It captures a broader range of practical problems compared to the traditional model. Although more motivations and examples of the application of such models are appreciated, this article paves the way for many future directions should it gather more research attention.

## Originality
Introducing noise into the query poses significant challenges for designing and analysing an appropriate algorithm. I did not read through the proofs in the appendix, but they seem technical and highly nontrivial.

---

> ### Author Rebuttal · Authors · 2026-03-27
>
> Thank you for the helpful feedback and suggestions. Some responses are as follows:
>
> **[Large number of items / more examples of real-life problems]**
>
> We would like to make several points addressing this:
> - We first note that Reviewer PCq3 asked a related question on scenarios where we might have noisy queries, and two examples can be found in the response to that reviewer.
> - Regarding your concern on the results holding for large $m$: Being the first study of the specific problem under consideration, we set our goals at the “challenging but realistic” level of establishing the sample complexity in $O(\cdot)$ notation as $m \to \infty$ and $\Delta$ varies with $m$ (typically $\Delta \to 0$).  We contend that establishing a precise non-asymptotic understanding for every finite $(m,\Delta)$ would be overly ambitious for an initial work.
> - Further discussion on applications: For context, we mention that the 2-agent setting has been widely considered in the literature, often for similar reasons of managing a problem’s difficulty (e.g., “Communication Complexity of Discrete Fair Division”, “Fair Division of Indivisible Items Between Two People with Identical Preferences”, “Almost Envy-Freeness with General Valuations”, and many more).  Regarding applications specifically having two agents and (potentially) many items, some are highlighted in Section 1.1.1 of “Communication Complexity of Discrete Fair Division”, including inheritance, border disputes, divorce settlements, and a real-world website called Fair Outcomes Inc.
>
> **[What if $\Delta$ is unknown and hence we can’t set $q$?]**
>
> Instead of explicitly “estimating $\Delta$”, we believe it would be better to answer this question from the viewpoint of “What if the algorithm can keep asking for more queries until it can certify that it has found an envy-free allocation?”, which achieves the same ultimate goal.
>
> As a warm-up, this can easily be done for the simple approach in Section 3.  That algorithm explicitly forms confidence intervals for the items, and more data can be collected to tighten them (with some extra care via a union bound) until they are tight enough to “certify” envy-freeness.
>
> For our main algorithm in Section 4, such “certification” is not immediate, but can be achieved using a method akin to “cross-validation”:
> 1. Run the algorithm with some initial number of queries $q_0$.
> 2. Perform the same queries a second time, but this time use them only for the purpose of estimating the envy.
> 3. Use a concentration argument to establish that the true envy lies within upper and lower limits (with the above estimate being halfway between these).  If the lower limit is $\le 0$, then we are confident that we have an envy-free allocation, and thus stop.
> 4. If not stopped, then we double the number of queries (i.e., first to $2q_0$ then subsequently $4q_0$, $8q_0$, etc.) and return to Step 1.
>
> We omit the detailed analysis of this procedure, since we believe it would be best not to add significant technical content to the paper post-submission.
>
> **[For known $\Delta$, given an instance, we don’t know the number of queries?]**
>
> We are not sure that we fully understood the question, but in the sense of “minimax guarantees”, which is our focus, knowing $\Delta$ and $m$ means that we *do* know from Theorem 2 that the number of queries $q$ should scale as $\frac{m^{2.5}}{\Delta^2}$ (up to logarithmic factors).  If the reviewer is thinking about more “instance-dependent” properties beyond $\Delta$ alone, then we acknowledge that this could indeed be of significant interest, but also likely highly challenging and firmly outside our current scope, and thus better left for future work.

---

> > ### Author Rebuttal · Reviewer_2nQd · 2026-04-01
> >
> > Thank the authors for the clarifications; most of my questions are addressed.
> >
> > Regarding Q3, I made a typo and was actually assuming $\Delta$ was unknown. So I was wondering whether the Algorithm in Section 4 consistently outperforms the Algorithm in Section 3 when they use the same number of queries.

---

> > > ### Author Response · Authors · 2026-04-02
> > >
> > > Thanks for the clarification.
> > >
> > > We believe more careful wording would be needed, in particular avoiding language like "always" and "consistently".  Comparing the two **upper bounds**, we see that the main algorithm has a significant reduction of $\sqrt{m}$.  However, there are some important caveats:
> > > - Comparing two upper bounds does not give a conclusive statement.  To do that, we would need to compare the main algorithm's upper bound to a **lower bound** for the simpler algorithm.  However, establishing algorithm-specific lower bounds was not a goal in our work.
> > > - The upper bounds hold in the minimax sense, i.e., worst-case instances.  As noted in our earlier reply, a detailed instance-dependent could reveal more subtle differences, though we expect this to be very challenging.
> > > - Even if the above caveats were fully resolved, a more carefully-worded statement might need to be to the effect of "if the simpler algorithm succeeds (e.g., success probability 0.99), then so does the main algorithm", or equivalently "if the main algorithm fails (e.g., success probability 0.01), then so does the simpler algorithm".  This would not rule out the possibility that, when $q$ is too small for both, the main algorithm fails more drastically (e.g., having success probability 0.0001 instead of 0.001).
> > >
> > > We would be happy to revise the paper to further convey that the limitation in Section 3 is **the overall combination of the simpler algorithm and its analysis**, rather than giving the impression that it's mainly/entirely the algorithm.

---

### Official Review · Reviewer_ZxxX · 2026-03-11

**Soundness:** 3
**Presentation:** 3
**Significance:** 3
**Originality:** 3
**Overall Recommendation:** 4
**Confidence:** 4

**Summary:**

The authors introduce the problem of envy-free allocation of indivisible goods when agent valuations can only be accessed through noisy (Gaussian) queries. They focus on a two-agent setting with additive utilities.  As part of their analysis, they show -

a. Upper and lower bounds:  They propose a polynomial-time algorithm to get the upper bound using non-adaptive queries and an item-by-item thresholding allocation rule.  Further, using multiple hypothesis testing and a randomized hard instance, they establish an algorithm-independent lower bound of $\tilde{\Omega}(\frac{m^{2.5}}{\Delta^{2}})$.
b. To make the problem tractable, they introduce the "gap" parameter $\Delta > 0$, representing how far the optimal allocation is from violating envy-freeness.
c. Their analysis extends beyond constant noise to general levels ($\sigma^2$), showing how the complexity shifts based on whether $\Delta$ is above or below $m^{3/4}$.

**Compliance With Llm Reviewing Policy:**

Affirmed.

**Key Questions For Authors:**

My main question is related to the $\Delta$ restriction.  Can anything from the current analysis be used for the developing algorithms for the $\Delta \ll m^{1/4}$ regime?  If there is no clear path to that regime, I feel the contributions of this work are somewhat diminished.

**Limitations:**

Yes.

**Strengths And Weaknesses:**

Strengths:

1. Optimal Scaling: The paper successfully closes the gap between the upper and lower bounds for the regime $\Delta \gg m^{1/4}$.
2. Addressing Suboptimality: The authors include a "naive" analysis (Section 3) that requires $O(m^3/\Delta^2)$ queries to highlight why their main algorithmic contribution, which uses bundle-level concentration rather than item-level intervals, is superior.
3. Practical Efficiency: Unlike brute-force approaches that check all possible allocations, the main algorithm runs in polynomial time.
4. Handling Few Queries: The analysis includes the "large $\Delta$" regime where $q < m$ (fewer queries than items), showing that sampling a subset of items can still achieve envy-freeness if $\Delta$ is sufficiently large.

Weaknesses:

1. The $\Delta$ Restriction: The main results require $\Delta \ge m^{1/4} \log^2 m$ which could be too large a value for negative envy. The authors acknowledge that the regime $\Delta \ll m^{1/4}$ likely requires a different class of algorithms that form bundles rather than using item-by-item thresholding.
2. Two-Agent Constraint: While the two-agent setting is a standard starting point in fair division, real-world applications often involve $n > 2$ agents. The authors state that this will require a non-trivial number of comparisons.
3. Gaussian Assumption: The upper bound analysis relies heavily on the specific properties of Gaussian noise. While the lower bound's use of Gaussian noise is a strength (showing hardness even in a standard model), the upper bound might require significant rework to generalize to all sub-Gaussian distributions.

---

> ### Author Rebuttal · Authors · 2026-03-27
>
> Thank you for the helpful feedback and suggestions. Some responses are as follows (mostly pointing to other reviewer responses where we address the same points):
>
>
> **[Weaknesses – $\Delta$ restriction, two agents, Gaussian noise]**
>
>
> The reviewer has indeed correctly identified the 3 main limitations of our work.  Because these are the most important limitations, we have paid attention to discussing them all carefully and outlining why they are challenging – the $\Delta$ restriction in Remark 2, the two-agent limitation in Section 7, and the Gaussian assumption in Remark 1.
>
>
> For more on the $\Delta$ restriction, please see the response to Reviewer PCq3.
>
>
> For more on the assumption of two agents, please see the first response to Reviewer 2nQd.
>
>
> **[Main question is related to the $\Delta$ restriction]**
>
>
> As noted above, we address this in the response to Reviewer PCq3.

---

> > ### Author Rebuttal · Reviewer_ZxxX · 2026-03-31
> >
> > I thank the authors for clarifying the two concerns I raised - the two-agent setting and the small-$\Delta$ regime.  I understand the hardness/challenges of addressing both these assumptions.  Having said that I feel these assumptions do restrict the magnitude of contributions of this work and therefore I'll maintain my original rating.

---

### Official Review · Reviewer_PCq3 · 2026-03-12

**Soundness:** 3
**Presentation:** 2
**Significance:** 1
**Originality:** 3
**Overall Recommendation:** 3
**Confidence:** 3

**Summary:**

This paper studies envy free allocation when agents’ valuations cannot be observed exactly and can only be accessed through noisy queries. The authors focus on a setting with two agents, additive valuations, and Gaussian noise in the query responses. The paper introduces the notion of an envy gap, which measures how close an allocation is to violating envy freeness, and shows that the number of queries needed depends on the size of this gap. The authors design a non adaptive querying strategy together with a simple threshold based allocation rule, and provide an upper bound on the number of queries required to find an envy free allocation. The paper also proves a lower bound that holds even for adaptive algorithms. Overall, the paper gives a theoretical analysis of fair allocation under noisy feedback and studies how the difficulty of the problem changes with the number of items and the envy gap.

**Compliance With Llm Reviewing Policy:**

Affirmed.

**Key Questions For Authors:**

1.	The main result does not cover the case where the envy gap becomes very small. Do the authors expect the same complexity to hold in this regime or would a different approach be required?
2.	Can the authors provide a concrete example that motivates the noisy query model? It would be helpful to understand in which real scenarios valuations are observed only through noisy feedback.

**Limitations:**

The paper also has several issues related to presentation and formatting. The related work section is quite long and takes up a large portion of the paper, which reduces the focus on the main technical results. The paper introduces many lemmas, some of which seem unnecessary for understanding the core contributions and make the presentation heavier than needed. In addition, some equations are followed by numbering that appears unnecessary.  Improving the structure and formatting would make the paper easier to read.

**Strengths And Weaknesses:**

Strength:
1.	The paper studies a novel variant of the fair division problem where agents’ valuations are accessed through noisy queries rather than exact information. This formulation captures an interesting extension of classical fair allocation models and raises nontrivial theoretical questions.
2.	The paper provides solid theoretical results. It establishes both upper and lower bounds on the query complexity and the bounds match up to logarithmic factors in a broad range of regimes.

Weakness:
1.	The main result does not cover the case where the envy gap is very small. Since the problem becomes harder as the gap decreases, leaving this case unresolved makes the theoretical results less complete.
2.	The paper does not clearly explain why agents' valuations would only be available through noisy queries. A concrete real-world example would help justify why this problem setting is important.
3.	The related work section takes a relatively large portion of the paper, which makes the presentation of the main results less focused. A more concise discussion of related work might help highlight the key technical contributions more clearly.

---

> ### Author Rebuttal · Authors · 2026-03-27
>
> Thank you for the helpful feedback and suggestions.  Some responses are as follows:
>
> **[Small-$\Delta$ case unresolved]**
>
>  Indeed, our upper and lower bounds provide a tight characterization for most regimes but not all regimes, as is **very common** in theory papers, particularly when studying a new problem.  In the area of bandit algorithms alone (which is closely related to our setup), here are just a few examples:
> - *Dueling bandits:* Initial bounds in “The K-armed Dueling Bandits Problem” (2012), optimal bounds in “Regret Lower Bound and Optimal Algorithm in Dueling Bandit Problem” (2015)
> - *Fixed-confidence pure exploration:* Initial bounds in in “PAC Bounds for Multi-armed Bandit and Markov Decision Processes” (2002), optimal bounds in “Optimal Best Arm Identification with Fixed Confidence” (2016)
> - *Rotting bandits:* Initial bounds in “Rotting Bandits” (2017), optimal bounds in “Rotting Bandits are not Harder than Stochastic Ones” (2019).
>
> For our particular problem, in the regime $\Delta \ll m^{1/4}$, we do get an upper bound from Section 3, but it is a $\sqrt{m}$ factor higher than the lower bound.  We would like to discuss various evidence for why a complete understanding may be highly challenging and thus very reasonable to leave for future work:
> - Firstly, we already carefully identified and discussed in Remark 2 where and why our algorithm from Section 4 fails, strongly suggesting that the regime $\Delta \ll m^{1/4}$ requires a different approach, or even an entirely different algorithm.
> - To elaborate on Remark 2, we would like to mention the following evidence of computational barriers for sufficiently small $\Delta$ values (though it is admittedly unclear how small): When $\Delta = 0$, the goal becomes finding an envy-free allocation with no further assumptions, and this problem is well known to be NP-hard (even with two agents and additive valuations) via a reduction from PARTITION.  Thus, it is very much feasible that sufficiently small $\Delta$ regimes face similar computational barriers.  Studying this would potentially need completely different techniques/ideas from our simple (and highly computationally efficient) algorithmic approach, and thus we believe it is very reasonable that we have left it as a gap for future research, especially since our current paper is already quite lengthy.
>
>
> **[Noisy queries and concrete real-world examples]**
>
> We would first like to point out that we have discussed the noise model’s motivations and limitations in detail in Remark 1. While our work is theoretical and not targeting a specific application, two potential examples are as follows:
>
> **(Example 1)** The “utilities” are not measurable or human-chosen, but instead need to be estimated via simulations that are stochastic in nature (e.g., estimating how much a particular resource will help each team in a company’s internal division problem).
>
> **(Example 2)** Each so-called “agent” is actually a group of people, and the “utility” is considered to be an average over the group members.  To query an item, we simply take a uniformly random group member and ask for their valuation.  (This formulation assumes that fairness is only required on an averaged group level, not an individual level.)
>
> Beyond the viewpoint of any particular application, we would also like to highlight the following:
> 1. The notion of noisy or uncertain valuations is gaining significant interest in the community on fair allocation and related problems, as we detailed in Section 1.1.
> 2. The study of algorithms that operate with noisy queries is a rich topic in Theoretical Computer Science, dating back to early works such as “Computing with Noisy Information” (1993) and recently gaining significant interest (e.g., “Instance-Optimality in the Noisy Value-and Comparison-Model” in SODA 2020, “Optimal Bounds for Noisy Sorting” in STOC 2023, and “Tight Bounds for Noisy Computation of High-Influence Functions, Connectivity, and Threshold” in COLT 2025).
>
> **[Related work could be more concise]**
>
> Thank you for the feedback.  We will consider moving some of the less related paragraphs to an appendix, but we view this as mainly a matter of taste, and in our experience a Related Work section slightly exceeding 1 page is not uncommon.
>
>
> **[Many lemmas and heavy presentation]**
>
> We intentionally placed all such lemmas in the technical appendices so that they would not distract from the high-level understanding being conveyed in the main text.  All of these lemmas are used carefully in our proofs, and we do not immediately see any alternative re-arrangement that would reduce how heavy the appendices are overall – the detailed proofs are simply highly technical in nature.  Having said this, we are open to suggestions.

---

### Decision · Program_Chairs · 2026-04-30

**Decision:**

Accept (regular)

**Comment:**

The paper introduces the problem of envy-free allocation of indivisible goods in which agents' values can be accessed only through noisy oracle queries. The paper studies a setting with two agents, and Gaussian noise and bounded valuations. As their main contribution, they give a tight result, closing the gap between upper and lower bound, on the optimal number of queries. Here, some amount of envy is acceptable, parameterized by $\Delta$. The paper shows its positive results when $\Delta \geq m^{1/4}\log^2 m$, and gives good reasons for why the smaller $\Delta$ regime requires newer ideas. Likewise, the setting beyond 2 agents also requires new ideas.

Overall, the paper's contribution of a tight result is a solid one, although its impact is limited by the rather strong restrictions like the large $\Delta$-regime and the setting of 2 agents. The paper squarely falls in the weak-accept regime: a good paper for ICML, but not a slam-dunk.